# TFAR: A Training-Free Framework for Autonomous Reliable Reasoning in Visual Question Answering

**Zhuo Zhi**  *zhuo.zhi.21@ucl.ac.uk*
*Department of Electronic and Electrical Engineering*
*University College London* [*]

**Chen Feng**  *chen.feng@ucl.ac.uk*
*Department of Electronic and Electrical Engineering*
*University College London*

**Adam Daneshmend**  *adam.daneshmend@nhs.net*
*Imperial College Healthcare NHS Trust*

**Mine Orlu**  *m.orlu@ucl.ac.uk*
*UCL School of Pharmacy*
*University College London*

**Andreas Demosthenous**  *a.demosthenous@ee.ucl.ac.uk*
*Department of Electronic and Electrical Engineering*
*University College London*

**Lu Yin**  *l.yin@surrey.ac.uk*
*School of Computer Science and Electronic Engineering*
*University of Surrey*

**Da Li**  *dali.academic@gmail.com*
*Samsung AI Centre Cambridge*
*Queen Mary University of London*

**Ziquan Liu**  *ziquan.liu@qmul.ac.uk*
*School of Electronic Engineering and Computer Science*
*Queen Mary University of London*

**Miguel Rodrigues**  *m.rodrigues@ucl.ac.uk*
*Department of Electronic and Electrical Engineering*
*University College London*

**Reviewed on OpenReview:** *https://openreview.net/forum?id=cBAKeZN3jy*

## Abstract

Recent approaches introduce chain-of-thought (CoT) reasoning to mitigate the challenges, such as hallucination and reasoning deficit in multimodal large language models (MLLMs) and enhance performance. However, existing CoT-based methods often rely on extensive data annotation and training. To overcome these limitations, we propose a training-free framework for autonomous and reliable reasoning (TFAR), which only uses common lightweight vision tools to improve the reasoning ability of MLLMs. TFAR enables an MLLM to autonomously and accurately identify relevant regions of interest (RoIs) and support CoT reasoning, without requiring additional training or annotations, and with low computational overhead during inference. However, the use of external tools will introduce

---

[*]Corresponding author

noise and uncertainty. To mitigate the uncertainty introduced by external tools and select the optimal pathway, we propose a conformal prediction-based uncertainty quantification method that calibrates the outputs from external tools and dynamically selects the most appropriate tool based on the MLLM's output uncertainty. Experiments across five datasets demonstrate that TFAR improves performance over the base MLLM by an average of 4.6%, in some cases even outperforming fine-tuned baselines, while maintaining low inference cost. These results offer new insights into training-free CoT guidance for MLLMs and underscore the value of reliable visual tools.

# 1 Introduction

To address common challenges in multimodal large language models (MLLMs)—such as hallucination (Wu et al., 2025) and reasoning deficit (Zhang et al., 2023)—recent studies have introduced chain-of-thought (CoT) reasoning to improve performance on tasks like visual question answering (VQA) (Shao et al., 2024; Xu et al., 2024; Zhi et al., 2025b). A central component of CoT in MLLMs is guiding the model from coarse-to-fine-grained visual understanding, gradually focusing attention on the most relevant image regions. This process reflects how humans typically interpret images: starting with a broad overview and then zooming in on specific details (Shao et al., 2024). For instance, (Shao et al., 2024) creates a dataset with bounding boxes highlighting question-relevant regions and designs a two-stage VQA pipeline. An MLLM fine-tuned on this data first identifies key regions, then generates answers by integrating both local and global cues, as illustrated in Fig. 1(a). Similarly, (Xu et al., 2024) introduced the LLava-O1-100K dataset, framing VQA at multiple granularities and employing stage-wise beam search. While these models show improved reasoning, they rely heavily on manual annotations and additional training costs that scale with model size and complexity.

Taking (Shao et al., 2024) as an example, a natural question arises: is it necessary to fine-tune an MLLM to teach it to select regions of interest (RoIs), as shown in Fig. 1(a) Could we instead prompt an off-the-shelf MLLM to select RoIs on its own, then combine these with the original image, and perform CoT reasoning without further training? To test this idea, we apply this naive two-round inference method using the setup from (Shao et al., 2024), without any additional tuning. As shown in Table 9, this approach fails to improve and often degrades overall accuracy. By analyzing the failure cases, we identify two key reasons: (1) Poor localization: the MLLM lacks a sense of scale, often misjudging image size, misplacing bounding boxes, or including irrelevant background; (2) Lack of instruction following: since the model is not trained to interpret prompts that request RoI selection, it tends to ignore such instructions and produces unreliable results, even format errors often occur. False RoIs arising from these factors can mislead the model's reasoning, ultimately resulting in incorrect answers. We show the representative failure examples in Fig. 6.

To address these limitations, we propose a training-free approach. We find that using only common lightweight vision tools (i.e., object detectors and segmenters) to provide essential visual information (i.e., the object name with bbox or boundary) can enable the MLLM to autonomously and accurately select RoIs by considering object size, position, and semantic relationships. In addition, with this information, MLLM only needs to select the object ID instead of generating a bounding box or boundary, thus avoiding problems such as format errors caused by inadequate instruction tuning. The selected regions are then combined with the full image to facilitate the final CoT reasoning by following the two-round reasoning in (Shao et al., 2024). Our approach promotes both deep visual understanding and precise RoI selection, without requiring any manual annotations or model retraining. A comparison with training-based methods is shown in Fig. 1.

However, using external tools introduces a new challenge: uncertainty and noise. Additionally, when multiple vision tools are available, it becomes unclear which one to trust. To address these issues, we incorporate uncertainty quantification (UQ) to evaluate both external tools and the MLLM itself, improving the reliability of the overall framework through uncertainty calibration or comparison. For example, Fig. 2 shows a case where an object detector misleads the MLLM and our proposed method based on UQ.

We integrate all the proposed methods into a framework named TFAR—Training-Free Framework for Autonomous Reliable Reasoning—as illustrated in Fig. 3. We follow (Shao et al., 2024) to use a two-stage

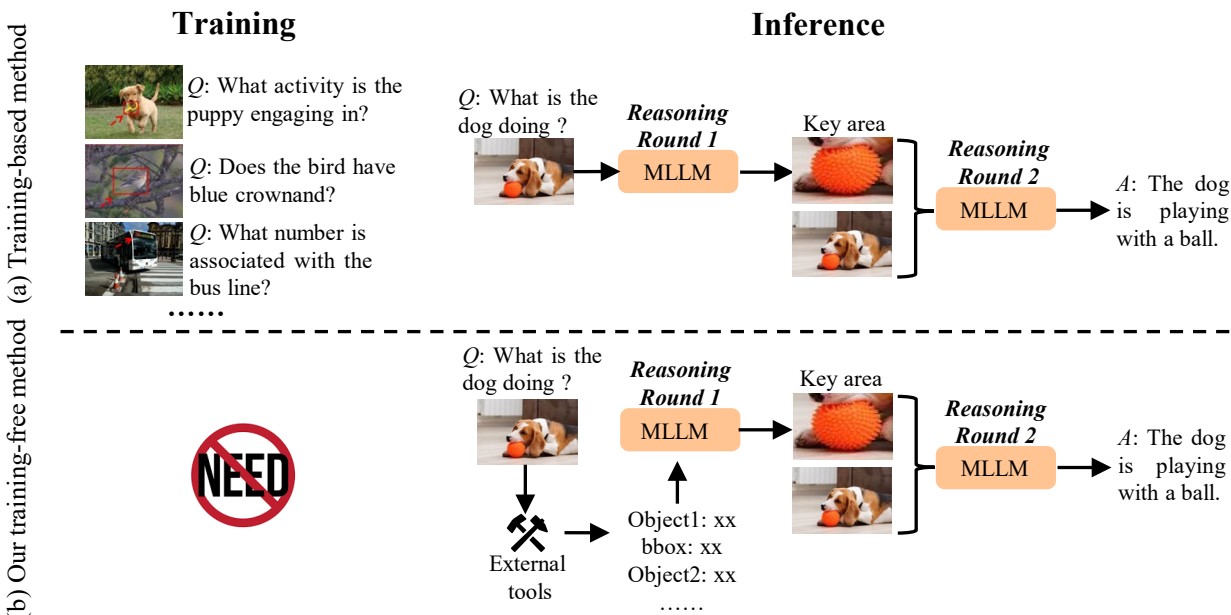

Figure 1: Comparison of training-based and our training-free methods for navigating CoT reasoning. **(a)** Training-based method (Shao et al., 2024): The MLLM is explicitly trained to identify key regions for CoT reasoning. A common approach involves constructing a dataset in which each image is annotated with regions relevant to the associated question, typically using bounding boxes. During training, the model learns to locate these regions, which are then extracted and used alongside the original image to support final-stage reasoning. While this strategy can enhance the reasoning performance of MLLMs, it requires extensive manual annotation and incurs significant training costs. **(b)** Our training-free method: Instead of training the MLLM to learn RoI selection, we only leverage lightweight vision tools to provide essential visual information (e.g., object names and bounding boxes). The MLLM then autonomously selects RoIs by reasoning over object size, position, and semantic relationships. These regions are combined with the original image to support final-stage CoT reasoning, without the need for manual annotations or fine-tuning.

process. The first stage of TFAR uses fine-grained visual information from external models, calibrated using a specially designed conformal prediction (CP)-based method (Vovk et al., 2005) for computational efficiency, to guide RoI selection. The MLLM uses this information, together with the query and image, to select relevant regions autonomously. In the second stage, the MLLM performs CoT reasoning based on both the original image and selected RoIs. To assess the reliability of this stage, we estimate uncertainty using a prediction set size-based method that considers the number of plausible tokens in top-$p$ sampling (Holtzman et al., 2020). This uncertainty is further used to select the most reliable vision tool for a given reasoning task. Our contributions are summarized as follows:

- We identify and address two key limitations in existing CoT-based multimodal reasoning pipelines—the need for extensive annotations and model retraining—by introducing a training-free reasoning framework. In this framework, using only common lightweight visual tools to provide basic information about the image enables MLLM to autonomously and accurately select RoIs, which facilitates the final CoT reasoning.

- We introduce a conformal prediction-inspired calibration strategy to quantify uncertainty from both external tools and MLLM outputs, ensuring that the two stages of MLLM inference are reliable. The strengths of the two UQ methods over existing UQ methods, such as confidence scores and entropy, are validated in our empirical study.

- We evaluate TFAR on five datasets, demonstrating that the performance of MLLM is significantly improved with an average improvement of 4.6%, which is on par with some methods that require additional training and annotations, while maintaining affordable computational cost. These findings

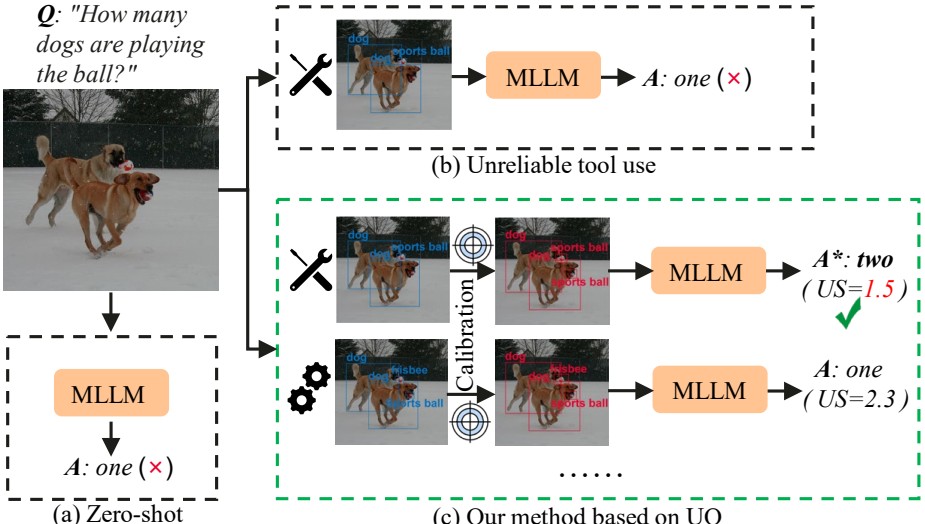

Figure 2: An example illustrating how unreliable visual tools can mislead MLLM reasoning. *US* refers to the uncertainty score. **(a)** Zero-shot VQA by the MLLM: The MLLM alone fails to answer correctly in zero-shot mode due to limited visual recognition capabilities. **(b)** MLLM with an unreliable object detector: Incorporating external visual information with noise can lead the MLLM to incorrect conclusions. **(c)** Our proposed method based on UQ: We calibrate the outputs of external vision tools to provide reliable visual cues and estimate the uncertainty of the MLLM's responses to identify the most reliable tool, which leads to the correct answer.

highlight the effectiveness of our proposed reasoning framework and the importance of incorporating visual tools reliably.

We review related work in Sec.2 and describe the proposed TFAR framework in Sec.3. Sec.4 presents the experimental settings and results, and we conclude our work in Sec.5.

## 2 Related work

**Vision language models** Vision language models integrate textual and visual representations to enable complex reasoning tasks and has always been the focus of much attention in the multimodal field (Zhang et al., 2024; Zhi et al., 2024; 2025a). Early approaches primarily focused on joint embedding strategies and attention mechanisms, where images and text are projected into a shared feature space for downstream tasks such as visual question answering, image captioning, and cross-modal retrieval (Lu et al., 2019; Kim et al., 2021). With the advent of LLM (Achiam et al., 2023; Touvron et al., 2023), researchers have explored ways to merge massive textual encoders/decoders with robust vision encoders, aiming to leverage LLMs' powerful linguistic reasoning while preserving rich visual features. One line of work investigates contrastive pre-training strategies, popularized by CLIP (Radford et al., 2021). Another family of methods explores fusion architectures—such as Flamingo (Alayrac et al., 2022) and BLIP-2 (Li et al., 2023)—where vision encoders are tightly coupled to transformer-based language models to achieve few-shot or even zero-shot performance on tasks ranging from image captioning to visual grounding. More recent efforts push the boundaries by aligning large language models with visual embeddings, named MLLM, such as LLaVA (Liu et al., 2024b), Qwen-VL(Bai et al., 2023) and InternVL (Chen et al., 2024). In our work, we mainly focus on incorporating training-free methods to improve MLLM reasoning in the VQA task.

**Chain-of-thought in visual reasoning.** The success of CoT in LLM is because it significantly improves the model's reasoning ability. With CoT, instead of answering questions directly, LLM simulates the reasoning of human beings and explicitly performs step-by-step reasoning to improve the performance in complex reasoning tasks (Wei et al., 2022; Zhang et al., 2022). Many works have begun to try to apply CoT in

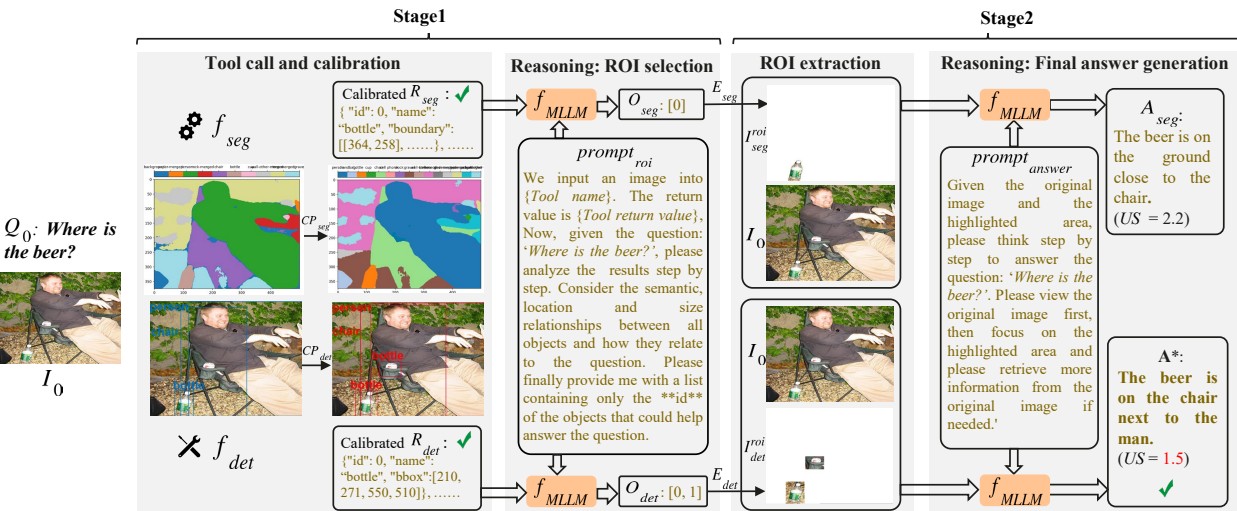

Figure 3: Overview of the proposed TFAR framework. When answering this example directly, MLLM will give the wrong answer: 'The beer is on the ground.'. Instead, this figure illustrates how TFAR generates the correct answer through a two-stage process. **Stage 1:** TFAR invokes external tools to obtain fine-grained information and applies CP-based calibration to their outputs to improve quality. This calibration mitigates issues such as the segmentation tool misclassifying many pixels as background and the object detection tool missing small objects. Based on the calibrated results, the MLLM autonomously selects RoIs through a reasoning process. **Stage 2:** The selected RoIs are extracted and combined with the original image as the MLLM input to perform the final CoT reasoning. The best answer is chosen from all pathways using our uncertainty estimation based on the prediction set size. TFAR does not require additional annotation or model training. By relying solely on common, lightweight vision tools to provide essential image information, the framework enables the MLLM to autonomously and accurately select RoIs while maintaining efficient inference. Uncertainty quantification at both stages further ensures the reliability and robustness of the entire reasoning process.

MLLM to enhance visual reasoning capabilities. Z. Z et al. (Zhang et al., 2023) propose a multimodal-CoT that incorporates language and vision modalities into a two-stage framework that separates rationale generation and answer inference. In this way, answer inference can leverage better generated rationales that are based on multimodal information. In (Shao et al., 2024), the authors create a dataset with bounding boxes for question-relevant regions in the image and design a two-stage VQA pipeline. An MLLM fine-tuned on this dataset automatically locates regions of interest and provides accurate answers by performing CoT with the help of global and local image details. Similarly, (Xu et al., 2024) introduces the LLavA-O1-100K dataset, which frames VQA at four levels of granularity and applies stage-level beam search. The resulting LLavA-O1 model displays strong reasoning capabilities. In addition, reinforcement learning is also used to guide the model to adaptively focus on key image areas, thereby achieving efficient multimodal reasoning (Zhang et al., 2025; Fan et al., 2025). However, these methods rely on extensive manual annotations and additional training, which pose practical challenges as MLLMs grow larger. In our approach, we show that only involving lightweight vision tools can enable the MLLM to autonomously and accurately select key regions, which facilitate the construction of the CoT. This removes the need for manual annotations and extra training.

**Uncertainty Quantification in MLLM** The quantification of uncertainty in LLMs has gained significant attention as a critical component for improving reliability and trustworthiness in AI systems (Xiong et al., 2023; Ye et al., 2025; Lin et al., 2023a). Some recent work has begun to extend it to MLLM. In (Kostumov et al., 2024), conformal prediction is employed to quantify the uncertainty of 20+ MLLMs and conclude that models with the highest accuracy may also have the highest uncertainty. However, they only focus on the multiple-choice VQA task. To solve the hallucination of MLLM under rare images, (Fang et al.,

2024) proposes to use uncertainty-guided token dropout to mitigate errors arising from visual token misinterpretation. Unlike in (Groot & Valdenegro-Toro, 2024), the verbalized uncertainty of MLLM is estimated via prompting and net calibration error is calculated to measure the direction of miscalibration. (Khan & Fu, 2024) proposes using the principle of neighborhood consistency to identify unreliable responses from an MLLM in QA tasks. Most of these approaches aim at black-box MLLM and do not consider uncertainty in using the external tools. In contrast, our TFAR framework estimates the uncertainty of both external tools and the MLLM's outputs, leading to more trustworthy results.

## 3 Method

In this section, we present the proposed methodology in two main sections: the proposed TFAR framework and the uncertainty quantification approach for it.

### 3.1 The proposed TFAR framework

We find that by referencing the basic information from external vision tools, MLLM can autonomously and accurately select RoIs without an additional learning process. These regions, combined with the original image, are subsequently fed into the MLLM. The MLLM then performs the final CoT reasoning, progressing from a coarse interpretation to fine-grained details, to generate the final answer. To ensure the reliability of both the vision models and the MLLM output, we utilize UQ for calibrating the tools and for selecting the final answer from multiple pathways. The overall framework is depicted in Fig. 3. For example, given an image $I_0$ and a question $Q_0$, the TFAR framework generates the final answer $A^*$ by following four main steps:

- **Tool call and calibration.** We call external tools to provide basic information of $I_0$, i.e., the object name and bbox/boundary from the segmentation tool $f_{seg}$ and object detection tool $f_{det}$ by following popular agent-based methods (Zhou et al., 2024; Wang et al., 2024). Note that in principle, TFAR can use any number of tools; here we use two tools as examples and show the results of more tools in the ablation study. To ensure the reliability of these tools, we design the calibration algorithms $CP_{seg}$ and $CP_{det}$ based on conformal prediction. Thus, we get the calibrated output of tools by $R_{seg} = CP_{seg}(f_{seg}(I_0))$ and $R_{det} = CP_{det}(f_{det}(I_0))$. Note that $R_{seg}$ and $R_{det}$ are in text format and objects are described using boundary point coordinates and bounding box, respectively.

- **Reasoning: RoI selection.** Instead of selecting the key area manually (Shao et al., 2024), We leave it to the MLLM $f_{MLLM}$ itself to choose the area that may help to answer the question based on the tool output $R_{seg}$ and $R_{det}$ by using the prompt $prompt_{roi}$: *'We input an image into {segmentation tool/object detection tool}. The return value is { $R_{seg}/R_{det}$}. Now, given the question $Q_0$, please analyze the results step by step. Consider the semantic, location and size relationships between all objects and how they relate to the question. Please finally provide me with a list containing only the \*\*id\*\* of the objects that could help answer the question.'*. In this process, MLLM reasons to fully explore the fine-grained information provided by tools and give the interested object id in the answer $O_{seg} = f_{MLLM}(R_{seg}, Q_0, prompt_{roi})$ and $O_{det} = f_{MLLM}(R_{det}, Q_0, prompt_{roi})$. Thanks to the basic information in $R_{seg}/R_{det}$, the MLLM only needs to select the object ID instead of generating a bounding box or boundary. This simplifies the task and helps avoid issues such as format errors that often arise from insufficient instruction tuning.

- **RoI extraction**. Based on the object of interest obtained in the previous step, we extract the corresponding region from the original image by $I_{seg}^{roi} = E_{seg}(I_0, O_{seg})$ and $I_{det}^{roi} = E_{det}(I_0, O_{det})$, where $E_{seg}$ refers to the area extraction by pixel for segmentation result and $E_{det}$ is the area extraction by bounding box for detection result.

- **Reasoning: Final answer generation**. We follow (Shao et al., 2024) to input the original image with the key area and guide the MLLM to perform CoT reasoning from coarse-grained to fine-grained by using the prompt $prompt_{answer}$: *'Given the original image and the highlighted area, please think step by step to answer the question {$Q_0$}. Please view the original image first, then focus on the*

*highlighted area and please retrieve more information from the original image if needed.'*. So far we get the final answer for both pathways by $A_{seg} = f_{MLLM}(I_0, Q_0, I_{seg}^{roi}, prompt_{answer})$ and $A_{det} = f_{MLLM}(I_0, Q_0, I_{det}^{roi}, prompt_{answer})$. In the end, we choose the more trustworthy answer as the final answer $A^* = \min_k US(A_k)$, $US$ is the proposed uncertainty score based on the quasi-conformal prediction that will be introduced in the next section.

Through the previous steps, TFAR provides a more trustworthy answer. We introduce the uncertainty-guided approach for TFAR in the next section.

## 3.2 Uncertainty quantification approach for TFAR

External tools can introduce uncertainty, which is a major challenge for TFAR. To extract trustworthy information from these tools and to select the most reliable pathway, we perform uncertainty quantification on both the tool return values and the MLLM outputs, using CP. CP is a distribution-free uncertainty quantification framework that provides statistically valid prediction sets with guaranteed coverage probability (Vovk et al., 2005). We describe our uncertainty quantification methods for external tools and the MLLM outputs, respectively.

### 3.2.1 Calibration of visual models by conformal prediction

Given a pre-trained model and an allowable error rate (or risk level) $\alpha$, the inductive CP constructs *prediction sets* that contain the true label with probability at least $1 - \alpha$, where $1 - \alpha$ is also named as the confidence level. In classification, one chooses a *nonconformity score* $s(\cdot)$ to measure how "unusual" each predicted label is (relative to the ground truth), then estimates a threshold $\hat{q}_\alpha$ from a held-out *calibration set*. Formally, let $\{(X_i, Y_i)\}_{i=1}^n$ denote a calibration set of size $n$, where $X_i$ is an input sample and $Y_i$ is its true label. Given a new, previously unseen test input $X_{\text{test}}$ and its (unknown) true label $Y_{\text{test}}$, conformal prediction produces sets

$$\mathcal{C}(X_{\text{test}}) = \{ \hat{y} \mid s(X_{\text{test}}, \hat{y}) \leq \hat{q}_\alpha \} \tag{1}$$

$$\text{s.t.} \quad \mathbb{P}\big(Y_{\text{test}} \in \mathcal{C}(X_{\text{test}})\big) \geq 1 - \alpha, \tag{2}$$

where $X_{\text{test}}$ denotes the test input and $Y_{\text{test}}$ denotes its (unknown) ground-truth label. The resulting set $\mathcal{C}(X_{\text{test}})$ is guaranteed to contain the true label with probability at least $1 - \alpha$. $\hat{q}_\alpha$ is the $(1 - \alpha)$-quantile of all calibration scores $\{s(X_i, Y_i)\}_{i=1}^n$ and is calculated by

$$\hat{q}_\alpha = \text{Quantile}\Big( \{ s(X_i, Y_i) \}_{i=1}^n \cup \{+\infty\}, \frac{\lceil (n+1)(1-\alpha) \rceil}{n} \Big). \tag{3}$$

The extra $\{+\infty\}$ and the ceiling term $\lceil (n+1)(1-\alpha) \rceil / n$ ensure finite-sample coverage guarantees (Angelopoulos & Bates, 2021). The core principle of CP is that choosing the quantile as shown in Equation 3 mathematically ensures that the prediction set of labels on the test data contains the true label with probability $1 - \alpha$. For a classification model whose output is a probability vector over classes, a common choice of nonconformity score is

$$s(X_i, Y_i) = 1 - p\big(Y_i \mid X_i\big), \tag{4}$$

i.e., the probability of the model not predicting the true class. Below, we show how CP can be specialized to (a) a segmentation model (pixel-wise) and (b) an object-detection model (bounding-box-wise).

**Calibration of a segmentation tool** Segmentation models can be viewed as a grid of pixel-level classifiers, each outputting a probability distribution over $K$ classes at every pixel location. A frequent practical issue is that many pixels belonging to foreground objects get labeled as *background* (class 0), causing under-segmentation (Xie et al., 2021). We calibrate the result using a pixel-wise CP approach that allows the re-labeling of certain background pixels. We define the following notions,

- $\mathcal{X}$ be the space of images of dimension $H \times W$;

- $\mathcal{Y} = \{0, 1, \ldots, K\}$ be the set of semantic labels (including background $= 0$);

- $\{(X_i, Y_i)\}_{i=1}^n$ be the calibration set, where $Y_i(u,v) \in \mathcal{Y}$ is the ground truth at pixel $(u,v)$;

- $N = n \times H \times W$ be the total number of pixels in the calibration set.

A segmentation model provides a probability vector $\mathbf{p}_{i,u,v} \in [0,1]^{K+1}$ over classes at each pixel $(u,v)$ in image $X_i$. We define the pixel-wise nonconformity score:

$$s(i,u,v) \; = \; 1 \; - \; p_{Y_i(u,v)}(i,u,v), \tag{5}$$

i.e., one minus the predicted probability that $(u,v)$ is its true class. Thus we can collect all nonconformity scores from the calibration set:

$$\mathcal{S} \; = \; \{\, s(i,u,v) \;\big|\; i = 1, \ldots, n; \; u = 1, \ldots, H; \; v = 1, \ldots, W \}. \tag{6}$$

Its $(1-\alpha)$-quantile $\hat{q}_\alpha$ (with finite-sample correction) is

$$\hat{q}_\alpha \; = \; \mathrm{Quantile}\Big(\, \mathcal{S} \cup \{+\infty\}, \; \frac{\lceil (N+1)(1-\alpha) \rceil}{N} \Big). \tag{7}$$

For a *test image* $X_{\text{test}}$, at each pixel $(u,v)$ we define the conformal prediction set

$$\mathcal{C}(u,v) \; = \; \{\, k \in \mathcal{Y} \;|\; 1 - p_k(u,v) \; \leq \; \hat{q}_\alpha \}. \tag{8}$$

Thus $\mathcal{C}(u,v)$ is all classes whose pixel-wise nonconformity scores are no larger than $\hat{q}_\alpha$. We map $\mathcal{C}(u,v)$ to a single *calibrated label* $\hat{y}_{\text{cal}}(u,v)$ via:

$$\hat{y}_{\text{cal}}(u,v) = \begin{cases} \arg \max\limits_{j \in \mathcal{C}(u,v)} p_j(u,v), & \text{if } \mathcal{C}(u,v) \neq \emptyset, \\ & and\ 0 \notin \mathcal{C}(u,v), \\ \arg \max\limits_{j \in \mathcal{C}(u,v) \setminus \{0\}} p_j(u,v), & \text{if } \mathcal{C}(u,v) \neq \emptyset, \\ & and\ 0 \in \mathcal{C}(u,v), \\ 0, & \text{if } \mathcal{C}(u,v) = \emptyset, \end{cases} \tag{9}$$

If 0 (background) appears in $\mathcal{C}(u,v)$, the pixel can be re-labeled to a foreground class within that set, mitigating over-conservative segmentation.

**Calibration of an object detection tool** We apply bounding-box-wise conformal prediction (Andéol et al., 2023) to calibrate object detections. This addresses extreme cases where tiny objects may require expanding the predicted box to guarantee covering the true object with high probability. We define the following notions: let

- $\mathcal{X}$ be the image space;

- $\mathcal{Y} = \{b^k\}_{k=1}^K$ be the set of $K$ ground-truth bounding boxes in an image, where each box is given by $b^k = (x_{\min}^k, y_{\min}^k, x_{\max}^k, y_{\max}^k)$;

- $\{(X_i, \{b_i^k\})\}_{i=1}^n$ be the calibration set, with $\{b_i^k\}$ the ground-truth boxes of image $X_i$.

A trained detector produces a set of predicted boxes $\{\hat{b}_i^j = (\hat{x}_{\min}^j, \hat{y}_{\min}^j, \hat{x}_{\max}^j, \hat{y}_{\max}^j)\}$ for image $X_i$. We match predicted boxes to ground-truth boxes via an IoU threshold $\tau$. Let $\mathcal{M}_i$ be the set of matched pairs $(\hat{b}_i^j, b_i^k)$ with $\mathrm{IoU}(\hat{b}_i^j, b_i^k) \geq \tau$. For each matched pair, define the *additive* nonconformity score:

$$\begin{aligned} s_j^k \; = \; \big(\, & \hat{x}_{\min}^j - x_{\min}^k, \; \hat{y}_{\min}^j - y_{\min}^k, \\ & x_{\max}^k - \hat{x}_{\max}^j, \; y_{\max}^k - \hat{y}_{\max}^j \big), \end{aligned} \tag{10}$$

which measures the coordinate-wise prediction errors. The full set of nonconformity scores is collected across all matched pairs:

$$\mathcal{S} = \bigcup_{i=1}^n \bigcup_{(\hat{b}_i^j, b_i^k) \in \mathcal{M}_i} \{ s_j^k[1], \; s_j^k[2], \; s_j^k[3], \; s_j^k[4] \}, \tag{11}$$

where $s_j^k[m]$ is the $m$th coordinate error. To ensure coverage on all four coordinates, we apply a Bonferroni correction, splitting risk $\alpha$ into $\alpha/4$ for each coordinate. Let $\mathcal{S}_m$ be the $m$th-coordinate errors across all matched pairs from all calibration images. Then for each $m \in \{1, 2, 3, 4\}$,

$$\hat{q}_{\alpha,m} \;=\; \text{Quantile}\Big(\mathcal{S}_m \cup \{+\infty\}, \;\; \frac{\lceil(|\mathcal{S}_m| + 1)\,(1 - \alpha/4)\rceil}{|\mathcal{S}_m|}\Big). \tag{12}$$

On a test image $X_{\text{test}}$, we conformalize each predicted box $\hat{b}^j$ by expanding it to

$$\begin{aligned}
\mathcal{C}(\hat{b}^j) \;=\; \Big[&\hat{x}_{\min}^j \;-\; \hat{q}_{\alpha,1}, \;\hat{y}_{\min}^j \;-\; \hat{q}_{\alpha,2}, \\
&\hat{x}_{\max}^j \;+\; \hat{q}_{\alpha,3}, \;\hat{y}_{\max}^j \;+\; \hat{q}_{\alpha,4}\Big].
\end{aligned} \tag{13}$$

This guarantees that if $\hat{b}^j$ is matched to a true box $b^k$, then $b^k$ remains inside $\mathcal{C}(\hat{b}^j)$ with probability at least $1 - \alpha$.

### 3.2.2 Uncertainty estimation for MLLM output based on prediction set

By using the TFAR framework, answers $A_{seg}$ and $A_{det}$ are generated through two pathways. We propose an uncertainty quantification method based on a prediction set in the top-$p$ sampling, corresponding to conformal prediction with a fixed non-conformity score threshold. Thus, we call it a quasi-conformal prediction UQ for convenience, but note that the top-$p$ sampling does not generate a prediction set with any coverage guarantee. This method measures the number of candidate tokens required to "cover" a predefined probability mass $p$ at each generation step in top-$p$ sampling. For an output sequence $A = \{w_1, w_2, \ldots, w_T\}$, where $w_i$ denotes the $i$th token, we compute the uncertainty score in two steps:

- Token probability sorting. For each token $w_i$, let $\mathcal{P}_i = \left[p_i^{(1)}, p_i^{(2)}, \ldots, p_i^{(V)}\right]$ represent its probability distribution over the vocabulary, sorted in descending order $\left(p_i^{(1)} \geq p_i^{(2)} \geq \cdots \geq p_i^{(V)}\right)$.

- Cumulative thresholding. We compute the minimal number of tokens $k_i$ needed to exceed a threshold $p \in (0, 1]$:

$$k_i = \min \left\{ k \;\bigg|\; \sum_{j=1}^{k} p_i^{(j)} \geq p \right\}. \tag{14}$$

  Here, $k_i$ reflects the breadth of plausible alternatives for $w_i$. A large $k_i$ indicates high uncertainty, while a small one implies confidence (few tokens dominate the probability mass).

- Sequence-level aggregation. The final uncertainty score (US) for $A$ is achieved by calculating the mean $k_i$ across all tokens:

$$US(A) = \frac{1}{T} \sum_{i=1}^{T} k_i. \tag{15}$$

By applying the proposed quasi-conformal prediction-based uncertainty estimation, we get the uncertainty score of two pathways $US(A_{seg})$ and $US(A_{det})$. The answer with a smaller uncertainty score will be considered as more trustworthy and thus output by the TFAR as the final answer $A^* = \min_k US(A_k)$

## 4 Experiment

We first describe the experimental settings, then the results of our method and the baselines across five multimodal reasoning datasets, demonstrating the effectiveness of our approach.

Table 1: Comparison with baselines on all datasets.

| Method | VQA2 | VizWiz | GQA | Flickr30K | MMBench | Average |
|---|---|---|---|---|---|---|
| LLaVA-1.5-13B | 80.0 | 53.6 | 63.3 | 62.3 | 67.7 | 65.4 |
| LLaVA-ov-7B (*base model*) | 79.0 | 53.6 | 64.8 | 61.1 | 72.3 | 66.2 |
| SPHINX-13B | 78.1 | 52.5 | 62.6 | 60.7 | 66.9 | 64.2 |
| vicuna-7B-VisCoT | 77.4 | 54.8 | 61.6 | 67.1 | 67.3 | 65.6 |
| **TFAR** (*ours*) | 82.5 | **60.1** | 67.3 | 67.2 | **77.1** | 70.8 |
| LLaVA-ov-7B-VisCoT (*upper bound*) | **83.3** | 57.2 | **69.6** | **69.9** | 76.6 | **71.3** |

## 4.1 Experimental setting

**Baselines.** We select the following strong baseline for comparison.

- Base model: LLaVA-OneVision-Qwen2-7b (LLaVa-OV) (Li et al., 2024). We selected this base model to build the TFAR pipeline due to its excellent performance and support for multiple image inputs, which is essential for our implementation. We refer to this baseline as LLaVA-ov-7B.

- LLaVA-1.5-13B (Liu et al., 2024a). LLaVA-1.5-13B is a popular baseline for VQA tasks with an increased parameter count.

- Visual CoT (VisCoT) (Shao et al., 2024). VisCoT is a state-of-the-art (SOTA) approach that fine-tunes an MLLM on a manually curated dataset, enabling it to automatically identify key areas in an image and facilitate CoT reasoning. We fine-tune the LLaVa-OV model on the VisCoT dataset following the procedure in (Shao et al., 2024) and denote the resulting model as LLaVA-ov-7B-VisCoT. Additionally, we also compare our method with the vicuna-7B-VisCoT model released in (Shao et al., 2024).

- SPHINX (Lin et al., 2023b). SPHINX enhances the MLLM's ability to identify regions of interest and guide CoT reasoning by incorporating multiple visual encoders during training. The base model used in SPHINX is Llama2-13B.

**Datasets.** We select a diverse set of VQA datasets to comprehensively evaluate our proposed method:

- VQA2 (Goyal et al., 2017): a large-scale benchmark for open-ended VQA tasks, featuring 107,391 question-answer pairs in the test set that cover diverse topics, including object recognition, counting, and commonsense reasoning.

- VizWiz (Gurari et al., 2018): a real-world dataset collected from blind or low-vision users, capturing everyday scenarios. The open-ended questions frequently involve tasks such as text recognition, object identification, and scene understanding, with 8,000 question-answer pairs in the test split.

- Flickr30K (Plummer et al., 2015): supports VQA with open-ended question-answer pairs that emphasize fine-grained object attributes, actions, and contextual scene understanding. The test set comprises 1,546 question-answer pairs.

- GQA (Hudson & Manning, 2019): provides 12,578 open-ended questions in the test set, with a focus on the relationships among objects, attributes, and spatial arrangements.

- MMBench (Liu et al., 2024c): includes 1,784 questions in the test set that span classification, captioning, and VQA tasks. It features both multiple-choice and open-ended questions across diverse visual scenarios.

**Metric.** All dataset evaluation scripts use accuracy as a metric, more details are given in the Appendix B.

**Tool selection.** We choose popular lightweight tools: SEEM (Zou et al., 2023) for segmentation and Yolov11 (Jocher & Qiu, 2024) for object detection. We also try other tools in the ablation study.

Table 2: Ablation of UQ approach in TFAR.

| Method | VQA2 | VizWiz | GQA | Flickr30K | MMBench | Average |
|---|---|---|---|---|---|---|
| base model | 79.0 | 53.6 | 64.8 | 61.1 | 72.3 | 66.2 |
| TFAR-seg-base | 80.3 | 56.2 | 65.2 | 62.3 | 74.0 | 67.6 |
| TFAR-det-base | 80.7 | 57.3 | 65.9 | 63.4 | 74.7 | 68.4 |
| TFAR-seg-CP | 82.0 | 59.0 | 66.6 | 66.1 | 75.5 | 69.8 |
| TFAR-det-CP | 81.7 | 58.6 | 66.2 | 65.2 | 75.8 | 69.5 |
| **TFAR (*ours*)** | **82.5** | **60.1** | **67.3** | **67.2** | **77.1** | **70.8** |

**TFAR setting** We set $\alpha = 0.1$ for both $CP_{seg}$ and $CP_{det}$, following prior work (Kumar et al., 2023; van der Laan & Alaa, 2025) and show more results in the Appendix C under different values of $\alpha$. Empirically, we set the parameter $p$ to 0.9 for the prediction set-based uncertainty estimation and show more results in the Appendix C under different values of $p$. To compute the nonconformity scores for both tools, we use the COCO-2017 validation dataset (Lin et al., 2014), which contains 5,000 images with annotations for both segmentation and object detection. We also try different calibration datasets in the ablation study. The time consumption for calculating the nonconformity score on the calibration set is 35 minutes and 8 minutes for the segmentation and object detection tools, respectively. For each test image, the average time taken for calibration is about 34ms and 0.2ms, respectively. The hardware and software platforms involved in the experiment are given in the Appendix B.

## 4.2 Main result

We present the experimental results of our proposed TFAR method alongside baselines on all datasets in Table 1. As shown in the table, TFAR substantially improves the base model's performance by 3.5%, 6.5%, 2.5%, 6.1%, and 4.8% across the five datasets, demonstrating its effectiveness. Notably, VizWiz exhibits the largest performance gain, likely because this dataset contains many blurred images, where TFAR's tool calibration plays a particularly important role. We show some calibrated results from the VizWiz dataset in Appendix C. Furthermore, TFAR outperforms all competing methods except for the upper bound, including some methods with more parameters (e.g., LLaVA-1.5-13B and SPHINX-13B). Remarkably, TFAR even surpasses the upper bound on VizWiz and MMBench, underlining the significance of reliable tool return values.

We show some representative examples in Fig. 4 to qualitatively analyze TFAR. Due to the limited space, We omit the interaction with MLLM in the figure and highlight the RoIs selection and UQ process. In Q1, the MLLM provides an incorrect answer due to interference from the cup pattern. The segmentation tool—especially after correction—accurately separates the cat from the cup, which helps the MLLM reach the correct answer with high confidence. Although the detection tool correctly identifies both the cup and the cat, its bounding box is too broad, failing to eliminate the interfering information. In Q2, the MLLM is confused by the image and produces an incorrect answer. In this case, both the segmentation and detection tools identify the dog; however, the segmentation tool also captures the person and refines the output through the calibration, which helps to better distinguish between the human and the dog and allows the MLLM to generate a more reliable answer. In Q3, MLLM fails to capture the bottle on the grass and gives an incomplete answer. The segmentation model and the detection model also face the same challenge. However, CP-based calibration enables the segmentation model to re-segment the bottle from the background, and the detection model successfully identifies this object. Consequently, MLLM focuses on the ball, the person, and the bottle, which leads to the correct answer. More visualizations are shown in the Appendix C.

## 4.3 Ablation study

We perform comprehensive ablation experiments to validate the effectiveness of the proposed approach.

**Inference efficiency** One potential concern with TFAR is inference efficiency, as it involves external tool calls and uncertainty calibration. To assess this, we compare the average inference time of TFAR with that

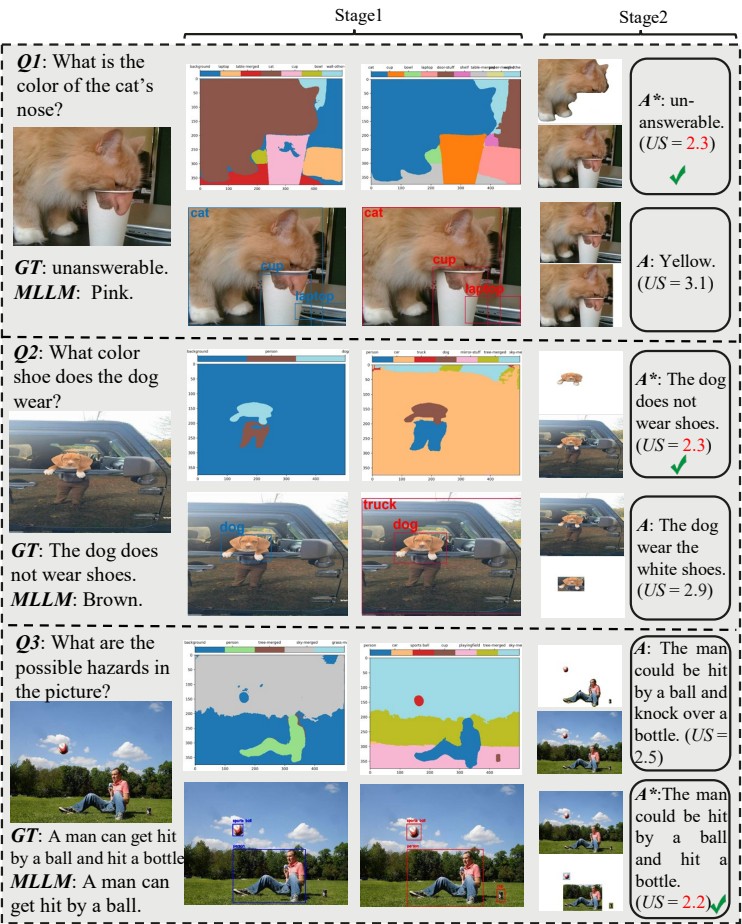

Figure 4: Visualization of some results. GT refers to the ground truth of the answer. Due to the limited space, We do not show interaction with MLLM, focusing on the RoIs selection and UQ process in Stage 1 and Stage 2. Refer to Fig. 3 for more details of the framework.

Table 3: Comparison of inference time consumption between LLaVA-ov-7B, TFAR and the LLaVA-ov-7B-VisCoT.

| Method | VQA2 | VizWiz |
|---|---|---|
| LLaVA-ov-7B | 1.24s | 1.33s |
| TFAR | 2.42s | 2.63s |
| LLaVA-ov-7B-VisCoT | 2.15s | 2.34s |

Table 4: Ablation of UQ approach in TFAR.

| Method | VQA2 | VizWiz | GQA | Flickr30K | MMBench | Average |
|---|---|---|---|---|---|---|
| base model | 79.0 | 53.6 | 64.8 | 61.1 | 72.3 | 66.2 |
| TFAR-VOC | 81.6 | 58.9 | 66.4 | 65.7 | 75.8 | 69.7 |
| **TFAR (*ours*)** | **82.5** | **60.1** | **67.3** | **67.2** | **77.1** | **70.8** |

of the best-performing baseline, LLaVA-ov-7B-VisCoT, and the base model, LLaVA-ov-7B, on the VQA2 and VizWiz datasets. All evaluations are conducted with a batch size of 1, and the average inference times across all samples are reported in Table 3. As shown in the table, TFAR's inference time is 12.6% and 12.4% slower than the baseline LLaVA-ov-7B-VisCoT on VQA2 and VizWiz, respectively. This performance overhead remains within an acceptable range, especially considering that LLaVA-ov-7B-VisCoT requires approximately 20 hours of fine-tuning with 438k question-answer pairs, which is computationally expensive and resource-intensive. The modest overhead introduced by TFAR primarily stems from the extended reasoning process in Stage 1, as demonstrated in Appendix C. Nevertheless, the integration of lightweight visual tools—each operating with millisecond-level latency—incurs minimal computational cost and does not introduce noticeable runtime delays (detailed in Section 4.1). Furthermore, the conformal prediction-based calibration step adds negligible overhead, as the inconsistency scores required for calibration are precomputed. It is worth noting, however, that the overall reasoning speed of both TFAR and VisCoT is substantially lower than that of the base model LLaVA-ov-7B. This reduction is attributed to the two-round dialogue framework employed by both methods to enhance reasoning quality.

**Different calibration dataset** Another important consideration is the impact of the calibration dataset distribution on overall performance. To investigate this, we evaluate TFAR using an alternative calibration set: the PASCAL VOC 2012 validation dataset (Everingham et al.), which contains 1449 images with both segmentation and detection annotations. We denote this variant as TFAR-VOC and compare its performance against the base model and the original TFAR (calibrated with the COCO 2017 validation set), as shown in Table 4. Our results show that TFAR-VOC exhibits a slight performance drop compared to TFAR, with an average decrease of 1.1%. Nonetheless, it still outperforms the base model by a substantial margin of 3.5%. We attribute this performance gap to the smaller number of samples and more limited category diversity in the PASCAL VOC dataset compared to COCO 2017. Based on these observations, we recommend using the COCO 2017 validation dataset for calibration to achieve the best overall performance.

**Different base models** We evaluate the effectiveness of TFAR when integrated with various base models, including both open-source and commercial MLLMs. The open-source models include mPLUG-Owl2 (7B) (Ye et al., 2024), and Qwen-VL (7B) (Bai et al., 2023). The commercial models include GPT-4o (Wang et al., 2024) and Gemini 2.5 Flash (Team et al., 2024). All of them support multi-image inputs. When combined with the TFAR framework, these are referred to as mPLUG-Owl2-TFAR, Qwen-VL-TFAR, GPT-4o-TFAR, and Gemini 2.5-TFAR, respectively. Note that the commercial models do not provide access to raw probability outputs. To address this, we follow a heuristic approach inspired by (Zhi et al., 2025b), allowing the LLM to produce its uncertainty score, in line with the concept of self-reflection (Ji et al., 2023). Specifically, we appended the following instruction to the original prompt: "Note that you need to generate an uncertainty score for your answer, scaled from 1.0 to 5.0. The larger the value, the more certain you are." . We show the detailed prompt in Appendix B, 'Prompt for the commercial models in stage 2'. As shown in Table 5, TFAR consistently improves the performance of all base models across five datasets, with average gains of 5.2%, 6.8%, 2.0%, and 1.3% for mPLUG-Owl2, Qwen-VL, GPT-4o, and Gemini 2.5, respectively.

Table 5: Comparison of different base models in TFAR.

| Method | VQA2 | VizWiz | GQA | Flickr30K | MMBench | Average |
|---|---|---|---|---|---|---|
| mPLUG-Owl2 | 78.3 | 49.8 | 56.1 | 58.5 | 64.5 | 61.4 |
| mPLUG-Owl2-TFAR | 80.3 | 56.9 | 62.6 | 63.8 | 69.4 | 66.6 |
| Qwen-VL | 79.5 | 42.6 | 59.3 | 56.0 | 60.6 | 59.6 |
| Qwen-VL-TFAR | 81.9 | 54.8 | 65.7 | 63.2 | 66.2 | 66.4 |
| GPT-4o | 89.1 | 72.2 | 83.3 | 79.4 | 83.4 | 81.5 |
| GPT-4o-TFAR | 89.8 | 76.7 | 85.1 | 80.5 | 85.4 | 83.5 |
| Gemini 2.5 | 91.4 | 75.3 | 86.2 | 84.1 | 86.9 | 84.8 |
| Gemini 2.5-TFAR | 91.9 | 78.8 | 86.9 | 85.3 | 87.7 | 86.1 |

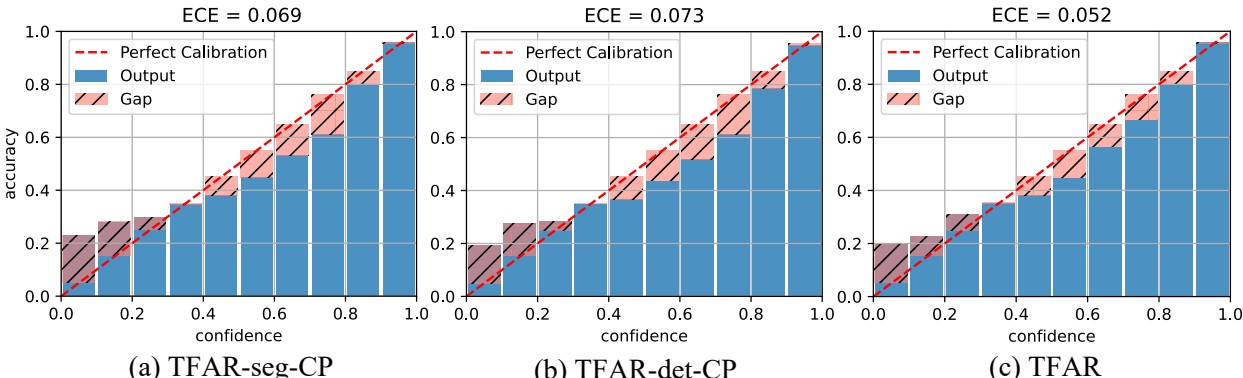

(a) TFAR-seg-CP      (b) TFAR-det-CP      (c) TFAR

Figure 5: Reliability diagram of TFAR-seg-CP, TFAR-det-CP and TFAR on GQA dataset.

These results demonstrate the generalizability of the proposed framework. Notably, the performance gains are more pronounced in open-source models. We attribute the smaller improvements observed in commercial models to their extensive pretraining on tasks such as object detection and segmentation, which equips them with strong visual understanding capabilities, sometimes surpassing those of dedicated vision models. Nevertheless, the significant performance boost observed in smaller open-source models underscores TFAR's practicality and potential in a wide range of application scenarios.

**Ablation of UQ approach in TFAR** We first compare the pipeline without using uncertainty to guide TFAR by disabling the calibration in Stage 1 and the uncertainty score calculation in Stage 2. This yields two pathways: TFAR-seg-base (segmentation tool) and TFAR-det-base (object detection tool). Next, we enable tool calibration in Stage 1 on top of TFAR-seg-base and TFAR-det-base to explore the benefit of the method, resulting in TFAR-seg-CP and TFAR-det-CP. The performances of these approaches are shown in Table 2. By comparing TFAR-seg-CP, TFAR-det-CP, and TFAR, it shows that using quasi-conformal prediction for answer selection improves the performance of a single pathway, and TFAR outperforms any single-pathway method on all datasets. Compared to the strongest single pathway, TFAR achieves an average improvement of about 1% on five datasets. This suggests that different samples require different vision models and that the uncertainty in the MLLM output can serve as a reliable metric for choosing the more appropriate tool.

We show the reliability diagram of TFAR-seg-CP, TFAR-det-CP, and TFAR on the GQA dataset in Fig. 5 to compare the reliability of the answer. A unique scores of 0 and 1 is calculated for each answer in the GQA dataset. We normalize the uncertainty scores of all answers before subtracting them from one to obtain the confidence score. It shows that the expected calibration error (ECE) (Guo et al., 2017) is reduced in TFAR, indicating the effectiveness of the proposed UQ for MLLM. Further comparison of TFAR-seg-base and TFAR-det-base with TFAR-seg-CP and TFAR-det-CP shows that CP-based tool calibration yields an improvement of 2.2% and 1.1% for two pathways, respectively. We attribute the higher performance of TFAR-seg-CP to its pixel-level ROI extraction, which excludes more irrelevant regions than box-level extraction and thus reduces the likelihood of introducing noise.

Table 6: Comparison of different ROI representation methods

| Method | VQA2 | VizWiz | GQA | Flickr30K | MMBench | Average |
|---|---|---|---|---|---|---|
| TFAR | **82.5** | 60.1 | **67.3** | **67.2** | **77.1** | **70.8** |
| TFAR-zoomin | 82.0 | **60.3** | 66.9 | 66.8 | **77.1** | 70.6 |

**Different UQ approaches in TFAR** We compare different UQ approaches. For the external tools, we compare CP with a heuristic method. In the segmentation result, the confidence score for each object is obtained by averaging the softmax probabilities of all pixels belonging to it. In the detection model, it uses the confidence score returned by the model. We refer to these two pathways as TFAR-seg-CS and TFAR-det-CS, respectively, and the overall approach as TFAR-CS. To incorporate the confidence score, we include it in the tool's return value (e.g., '{ *"id": xx, "name": 'xx', "confidence score": xx, "boundary": xx, ......* }') and add the text to $prompt_{roi}$: *'xxx, Consider the semantic, location, size and confidence relationships between all objects and how they relate to the question. xxx'*. For MLLM's output, we compare with uncertainty estimated by entropy:

$$US_{\text{entropy}}(A) = \frac{1}{T} \sum_{i=1}^{T} \left( - \sum_{j=1}^{V} p_i^{(j)} \log p_i^{(j)} \right). \tag{16}$$

We denote this approach as TFAR-US_entropy, where the tool calibration still uses CP. Table 7 presents the results. From Table 7 we see that heuristic-based uncertainty estimation for the tools underperforms the CP-based approach in TFAR, with drops of 1.6% for the segmentation pathway and 0.8% for the detection pathway. This effect is more pronounced in segmentation, likely because the heuristic struggles to handle overly conservative segmentation predictions. We also observe that using entropy for the MLLM output boosts performance by 0.4% compared to the strongest single pathway, but it still trails the quasi-conformal prediction approach by 0.6%. We attribute it to UQ by top-$p$ coverage focusing on major contributors to the probability mass and aligns more closely with practical uncertainty perceptions, whereas entropy factors in the entire long-tail distribution and can artificially inflate perceived uncertainty in some cases.

**More tools in TFAR** In principle, TFAR can be extended to include an arbitrary number of external tools. To explore the impact of incorporating additional tools, we integrate Mask2Former (Cheng et al., 2022) as a second segmentation model (denoted as TFAR-seg2) and InternImage (Wang et al., 2023) as a second detection model (TFAR-det2). The extended version of the framework is referred to as **TFAR+**. Experimental results are presented in Table 8. We observe that both TFAR-seg2-CP and TFAR-det2-CP perform comparably to their original counterparts. TFAR+ yields a modest performance improvement of 0.5%. This result suggests that the MLLM can already identify accurate regions of interest using the essential visual information provided by the initial two tools, without requiring additional computational overhead. These findings highlight that TFAR achieves strong performance with a minimal toolset, reinforcing its efficiency and scalability for practical deployment. Therefore, considering the trade-off between performance and efficiency, we recommend using the original TFAR configuration with two tools.

**Different ROI representation method in TFAR** In TFAR, we mask out other objects that are not of interest. Here, we use different ROI representation methods where, after the MLLM selects objects of interest, we magnify those objects. Specifically, we enlarge each object to the original image size. To avoid overlap among multiple objects of interest, we input all magnified images as separate images alongside the original image into the final MLLM. We refer to this approach as TFAR-zoomin, and evaluated it on all datasets. The results are shown in Table 6. Our results indicate that TFAR-zoomin performs slightly better than or comparable to TFAR on some datasets (e.g., VizWiz and MMBench), but overall, its performance is slightly lower than TFAR's. We hypothesize that magnifying objects in the middle of the image may disrupt the original spatial relationships, potentially misleading the MLLM's final judgment. Based on these findings, we believe that the original TFAR settings, which maintain spatial consistency, remain preferable.

Table 7: Comprison of different UQ approaches.

| Method | VQA2 | VizWiz | GQA | Flickr30K | MMBench | Average |
|---|---|---|---|---|---|---|
| TFAR-seg-CP | 82.0 | 59.0 | 66.6 | 66.1 | 75.5 | 69.8 |
| TFAR-det-CP | 81.7 | 58.6 | 66.2 | 65.2 | 75.8 | 69.5 |
| TFAR-seg-CS | 80.9 | 57.1 | 65.8 | 63.0 | 74.3 | 68.2 |
| TFAR-det-CS | 81.0 | 57.6 | 66.4 | 63.4 | 75.0 | 68.7 |
| TFAR-CS | 81.9 | 58.3 | 67.2 | 64.0 | 76.0 | 69.5 |
| TFAR-US_entropy | 82.0 | 59.4 | 66.9 | 66.5 | 76.3 | 70.2 |
| TFAR | **82.5** | **60.1** | **67.3** | **67.2** | **77.1** | **70.8** |

Table 8: Performance of TFAR+ with 4 tools

| Method | VQA2 | VizWiz | GQA | Flickr30K | MMBench | Average |
|---|---|---|---|---|---|---|
| TFAR-seg-CP | 82.0 | 59.0 | 66.6 | 66.1 | 75.5 | 69.8 |
| TFAR-det-CP | 81.7 | 58.6 | 66.2 | 65.2 | 75.8 | 69.5 |
| TFAR-seg2-CP | 81.5 | 58.6 | 66.0 | 65.7 | 75.0 | 69.4 |
| TFAR-det2-CP | 81.9 | 59.1 | 66.5 | 65.6 | 76.7 | 70.0 |
| TFAR | 82.5 | 60.1 | 67.3 | 67.2 | 77.1 | 70.8 |
| TFAR+ | **83.2** | **60.4** | **67.6** | **67.8** | **77.4** | **71.3** |

## 5 Conclusion

In this work, we introduced TFAR, a training-free framework for autonomous and reliable reasoning in the visual question answering (VQA) task. Specifically, we demonstrate that using only common, lightweight vision tools to extract essential image information enables multimodal large language models (MLLMs) to autonomously and accurately select regions of interest (RoIs), thereby facilitating effective chain-of-thought (CoT) reasoning. This approach eliminates the need for manual annotation and model fine-tuning. To enhance the reliability of the framework, we incorporate uncertainty quantification (UQ) based on conformal prediction (CP) at both stages of the reasoning process. The CP-based UQ not only improves the robustness of external visual tool outputs but also guides the selection of the most credible reasoning pathway. Extensive experiments on five diverse datasets—VQA2, VizWiz, Flickr30K, GQA, and MMBench—demonstrate that TFAR consistently improves performance over the base MLLM, with an average gain of 4.6%. In some cases, TFAR even outperforms methods that require additional training. Ablation studies on inference efficiency show that TFAR achieves this performance improvement with only a 12.5% increase in inference time over the best training-based baseline, offering a favorable trade-off that avoids the need for dozens of hours of model training and data annotation. Additional ablations confirm the effectiveness of our CP-based UQ strategy, which outperforms conventional heuristic-based approaches. In future work, we plan to extend TFAR to support additional modalities and tasks beyond VQA. We also consider training TFAR to enable the MLLM to autonomously select more suitable tools for a given task, thereby reducing the inference burden.

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

# Appendix

## A  Method

We summarize the proposed TFAR framework in Algorithm 1.

---
**Algorithm 1** The Proposed TFAR Framework

---
**Input:** Image $I_0$, Question $Q_0$, external tools $f_{seg}$, $f_{det}$, pre-trained MLLM $f_{\text{MLLM}}$, CP-based calibration methods ($CP_{seg}$, $CP_{det}$), the method for UQ of MLLM output $US$, pixel-based image extraction $E_{seg}$ and bounding box-based image extraction $E_{det}$.

    **Stage 1: Tool Call and Calibration**

1: Call segmentation tool: $R_{seg} \leftarrow CP_{seg}(f_{seg}(I_0))$

2: Call detection tool: $R_{det} \leftarrow CP_{det}(f_{det}(I_0))$

3: Select RoI via MLLM-CoT:

$$O_{seg} \leftarrow f_{MLLM}(R_{seg}, Q_0, prompt_{roi}),$$
$$O_{det} \leftarrow f_{MLLM}(R_{det}, Q_0, prompt_{roi})$$

4: Extract RoI images:
$$I_{seg}^{roi} = E_{seg}(I_0, O_{seg}), \quad I_{det}^{roi} = E_{det}(I_0, O_{det})$$

    **Stage 2: Final Answer Generation and Selection**

5: Perform final reasoning with CoT:

$$A_{seg} \leftarrow f_{MLLM}(I_0, Q_0, I_{seg}^{roi}, prompt_{answer}), \quad A_{det} \leftarrow f_{MLLM}(I_0, Q_0, I_{det}^{roi}, prompt_{answer})$$

6: Compute uncertainty scores:

$$US_{seg} = US(A_{seg}), \quad US_{det} = US(A_{det})$$

7: Choose the final answer with minimal uncertainty:

$$A^* = \arg\min_{A_k \in \{A_{seg}, A_{det}\}} US(A_k)$$

8: **return** Final Answer: $A^*$

---

## B  Experimental Settings

**Metric calculation method** All datasets involved in the paper use accuracy as an evaluation metric and calculate it as follows:

- VQA2. Each question in VQA2 is associated with 10 human-provided answers. For a predicted answer, the score is defined as: score = min(1, (number of matching human answers) / 3), which means if at least 3 annotators agree with the predicted answer, it receives full credit (a score of 1); fewer matches yield partial credit. The overall accuracy is the average of these per-question scores across the dataset.

- VizWiz. Similar to VQA2, VizWiz collects multiple answers per question (typically reflecting the diversity in responses from blind users). It uses essentially the same formula as VQA2—comparing the predicted answer against the set of human responses using the min(1, count/3) rule. The per-question scores are averaged to produce the final accuracy score.

- GQA. For each question, the answer is deemed correct if it exactly matches the ground truth provided by the dataset. The primary measure is the percentage of questions answered correctly (i.e., the

number of correct answers divided by the total number of questions). Beyond simple accuracy, GQA papers also explore secondary metrics (such as consistency, validity, and reasoning subtasks), but the headline metric remains overall accuracy.

- Flickr30K. The evaluation of answers in Flickr30K follows (Shao et al., 2024). Reference answers are provided, and predicted answers are evaluated against these references using a GPT-based scoring method. The final performance metric is computed by averaging the scores across all samples.

- MMbench. MMBench is designed to evaluate multimodal models over a range of tasks. For each question or task, answers are scored on a scale (often normalized between 0 and 1 or 0 and 100) based on criteria like correctness, completeness, and relevance. The scoring can be performed using human evaluations, automated assessments (e.g., leveraging language models as evaluators), or a combination of both. Each answer receives a score reflecting how well it meets the task requirements. These per-task (or per-question) scores are then averaged to yield a composite performance metric that summarizes the model's overall capability across the diverse tasks included in MMBench.

**Prompt for the commercial models in stage 2** The commercial models do not provide access to raw probability outputs. To address this, we follow a heuristic approach inspired by (Zhi et al., 2025b), allowing the LLM to produce its uncertainty score, in line with the concept of self-reflection (Ji et al., 2023). We also added a JSON template to facilitate parsing answers and uncertainty scores from MLLM responses. The complete prompt is as follows:

```
'''
Given the original image and the highlighted area, please think step by step
    ↪ to generate the answer [ANSWER] of the question Q. Please view the
    ↪ original image first, then focus on the highlighted area and please
    ↪ retrieve more information from the original image if needed.
    ↪ Additionally, you need to give me [UNCERTAINTY SCORE] that indicates
    ↪ your uncertainty in answering the question accurately, on a scale from
    ↪ 1.0 to 5.0  and accurate to one decimal place. The larger the value, the
    ↪  more certain you are. YOU MUST output in the JSON format:
{
'ANSWER': [ANSWER],
'UNCERTAINTY SCORE':  [UNCERTAINTY SCORE]
}
'''
```

**Experimental platform** All the experiments run on NVIDIA L40 GPU with CUDA 12.1.

## C   More experimental results

**Directly let MLLM select RoI and reason with the original image** We explore not training the MLLM but letting it directly select the region of interest and then perform VQA by following the two-round reasoning framework of VisCoT(Shao et al., 2024). In reasoning round 1, we use the prompt (abbreviation) '*Please think step by step and give the bbox of the area in this picture that may help answer the question, and answer in the format of [xmin, ymin, xmax, ymax]. If there are multiple regions, please provide multiple bboxes.*'. In reasoning round 2, we extract the area according to the bbox in the previous answer and combine it with the original image for final answer generation by using the prompt '*Given the original image and the highlighted area, please think step by step to answer the question. Please view the original image first, then focus on the highlighted area, and please retrieve more information from the original image if needed*'. We use LLaVA-OneVision-Qwen2-7b as the base model (because the model can handle multiple image inputs). The method of fine-tuning using the VisCoT dataset to facilitate localization of RoIs is called LLaVA-ov-7B-VisCoT. We name our approach as LLaVA-ov-7B-VisCoT-naive. We test all methods on the VQA2 and Flickr30K datasets, using accuracy as the metric. The results are shown in Table 9. As Table 9 shows, the results not only fail to improve but often degrade overall accuracy. By analyzing the failure cases, we

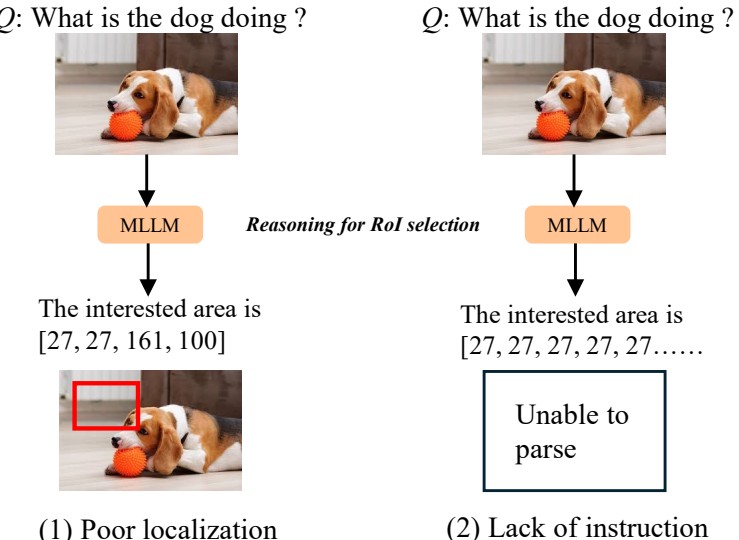

Figure 6: Failure cases of prompting MLLMs directly for RoI selection. (1) Poor localization. The MLLM lacks an understanding of absolute scale. It often misjudges the image size, resulting in bounding boxes that either miss the target or include excessive background. (2) Lack of instruction tuning for region selection. Since the model was not trained with prompts focused on identifying key regions, it tends to treat such instructions as secondary. This leads to unreliable region proposals and frequent format errors.

summarize two reasons: (1) Poor localisation. The MLLM has no sense of absolute scale. It often guesses the image size incorrectly, draws bounding boxes that miss the target, or includes too much background. (2) No instruction tuning for "find the key region." Because the model was never trained on prompts that emphasise region selection, it treats the request as a side note and delivers unreliable RoIs, and even format errors often occur. False RoIs arising from these factors can mislead the model's reasoning, ultimately resulting in incorrect answers. We show the representative failure examples in Fig. 6.

Table 9: Comparison with baselines on all datasets.

| Method | VQA2 | Flickr30K |
|---|---|---|
| LLaVA-ov-7B (*base model*) | 79.0 | 61.1 |
| LLaVA-ov-7B-VisCoT-naive | 78.4 | 59.2 |
| LLaVA-ov-7B-VisCoT | 83.3 | 69.9 |

**Visualization of more results** We show more representative examples in Fig.7. In Q1, the MLLM fails to distinguish between the bowling ball and the person's head, leading to an incorrect answer. Initially, the segmentation tool identifies only a portion of the person's region. However, after applying CP, it successfully segments both the bowling ball's and the head's regions. This improvement enables the model to capture the head's details and mitigate the bowling ball's interference, ultimately guiding the MLLM to the correct answer, whereas the detection model is unable to resolve the issue. In Q2, note the airplane toy includes a ***small captain toy on its front***—a detail that the MLLM initially missed. Although neither the segmentation nor the detection model recognized this feature at first, the calibrated detection tool eventually identified an object in that place, thereby preventing the model from focusing on the little girl, which assists the MLLM in arriving at the correct answer. In Q3, the MLLM failed to distinguish between the human and the dog, resulting in an incorrect answer. The segmentation tool only segments the dog while a large area of the image is labeled as background. In contrast, the detection tool identifies both the human and the dog, though its detection frame lacked precision. After CP calibration, the detection tool's output changed little, but the segmentation model's calibrated results successfully separated the human and the dog, enabling the MLLM to reach the correct answer with high confidence. In addition, combining Fig. 4

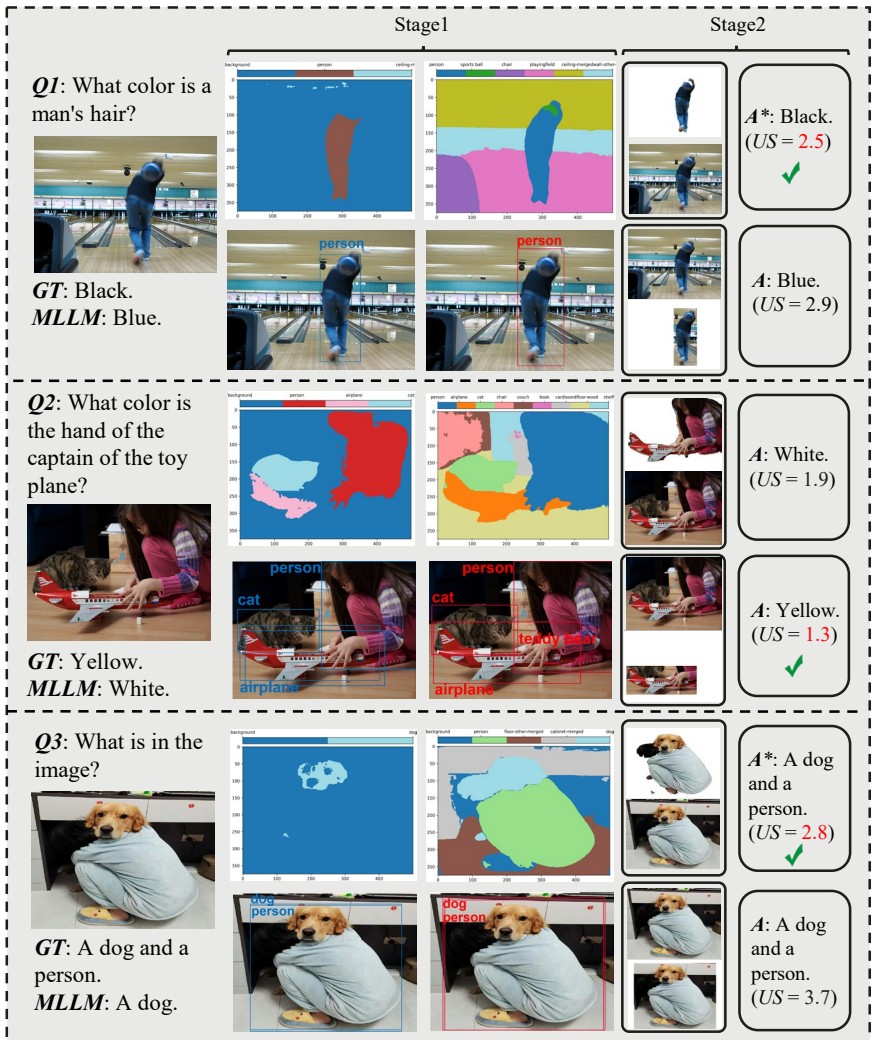

Figure 7: Visualization of more results.

and Fig. 7, we observe that the segmentation model performs better with overlapping targets, whereas the bounding boxes provided by the detection model struggle to accurately separate these targets. **Different $\alpha$ and $p$ in TFAR framework** We evaluate the effects of parameters $\alpha$ and $p$ in the TFAR framework by conducting experiments with varying settings. Specifically, we examine $\alpha$ values at 0.05, 0.1, 0.15, and 0.2, and $p$ values at 0.8, 0.85, 0.9, and 0.95. The performance results are summarized in Table 10 and Table 11. From Table 10, we observe that extreme values of $\alpha$ negatively impact the performance. Specifically, very small $\alpha$ values impose stringent error-rate constraints, resulting in wider, overly conservative prediction intervals that lack granularity and informativeness. Conversely, large $\alpha$ values permit higher error rates, yielding narrower, more precise intervals but significantly increasing the likelihood that the true value falls outside the interval. Table 11 indicates that TFAR performance remains relatively stable when $p$ is set to 0.95. However, reducing $p$ to 0.85 or 0.8 leads to noticeable performance degradation. This decline likely occurs because a lower threshold prematurely discards predictions with closely matched probabilities, thereby truncating the probability distribution and misrepresenting the actual uncertainty. Based on our analysis, we recommend selecting $\alpha = 0.1$ and $p = 0.9$ for optimal performance of TFAR.

**Comparison of reasoning length between TFAR and VisCoT** To help readers better understand the difference in inference processes between TFAR and VisCoT, we provide the average number of output tokens in the two reasoning rounds on the VQA2 dataset, as shown in Table 12. The primary difference

Table 10: The performance of TFAR with different $\alpha$.

| $\alpha$ | VQA2 | VizWiz | GQA | Flickr30K | MMBench | Average |
|---|---|---|---|---|---|---|
| 0.05 | 82.1 | 59.5 | 67.0 | 66.6 | 77.0 | 70.4 |
| 0.1 (*used in paper*) | **82.5** | **60.1** | **67.3** | **67.2** | **77.1** | **70.8** |
| 0.15 | 82.3 | 59.8 | 66.8 | 66.9 | 76.5 | 70.5 |
| 0.2 | 82.0 | 59.2 | 66.6 | 66.5 | 76.2 | 70.1 |

Table 11: The performance of TFAR with different $p$.

| $p$ | VQA2 | VizWiz | GQA | Flickr30K | MMBench | Average |
|---|---|---|---|---|---|---|
| 0.8 | 81.8 | 59.0 | 66.5 | 65.9 | 76.2 | 69.9 |
| 0.85 | 82.0 | 59.3 | 66.8 | 66.6 | 76.7 | 70.3 |
| 0.9 (*used in paper*) | **82.5** | **60.1** | **67.3** | **67.2** | **77.1** | **70.8** |
| 0.95 | 82.5 | 59.8 | 67.0 | 67.0 | 77.0 | 70.7 |

in CoT (Chain-of-Thought) length between TFAR and VisCoT occurs in the first stage. We illustrate this using the example in Fig. 8. In stage 1, TFAR explicitly prompts the MLLM to consider the semantics, size, and spatial relationships of all detected/segmented objects in the current image and select the index of the object of interest. In contrast, VisCoT has been trained on millions of examples to rapidly locate regions of interest, resulting in a shorter CoT in the first round. This difference explains why TFAR is slightly less efficient during inference. In stage 2—when generating the final answer based on the original image and the selected region of interest—the CoT lengths for both methods are comparable, as they receive similar image inputs and prompts.

**Visualization of detection failure and detection error of tools.** Based on our analysis of the experimental results, we observe that when a tool fails to detect any relevant information, it is often due to the image being inherently complex, blurry, or ambiguous. In such cases, the MLLM's own ability to understand the image is also limited. These failure cases can generally be categorized into two types: (1) Detection failure – where the tool does not return any useful information. In this situation, we fall back to using the original image as the input region of interest for the MLLM, essentially bypassing the tool. Whether the MLLM can still provide a correct answer depends entirely on its own visual reasoning capability. For example, in Fig. 9, the object detector failed to identify any objects, yet the MLLM was still able to answer correctly. (2) Detection error – where the tool returns incorrect or misleading information. In most of these cases, the MLLM is negatively influenced by the tool's output, highlighting the importance of tool correction. While we provide examples of such failure cases in Fig. 4, 7 and 2, we also include a success case in which the MLLM correctly rejects the tool's faulty output in Fig. 10. However, such outcomes are rare. When the tool produces incorrect results, it often indicates that the image is particularly challenging, making it difficult for the MLLM to compensate for the error on its own.

**Some calibrated results from VizWiz dataset.** The VizWiz dataset simulates images captured from the perspective of visually impaired individuals and primarily contains blurry or low-quality images. MLLMs generally exhibit limited robustness when processing such inputs directly. In TFAR, dedicated object detection and segmentation tools can effectively assist in identifying these challenging regions, thereby improving overall accuracy. To better illustrate these results, we show some results here. As shown in Fig. 11, the VizWiz dataset is characterized by frequent motion blur and poor image quality, which present significant

Table 12: Comparison of output token number in two stages between VisCoT and TFAR

| Method | Stage1 | Stage2 |
|---|---|---|
| TFAR | 132 | 115 |
| VisCoT | 14 | 132 |

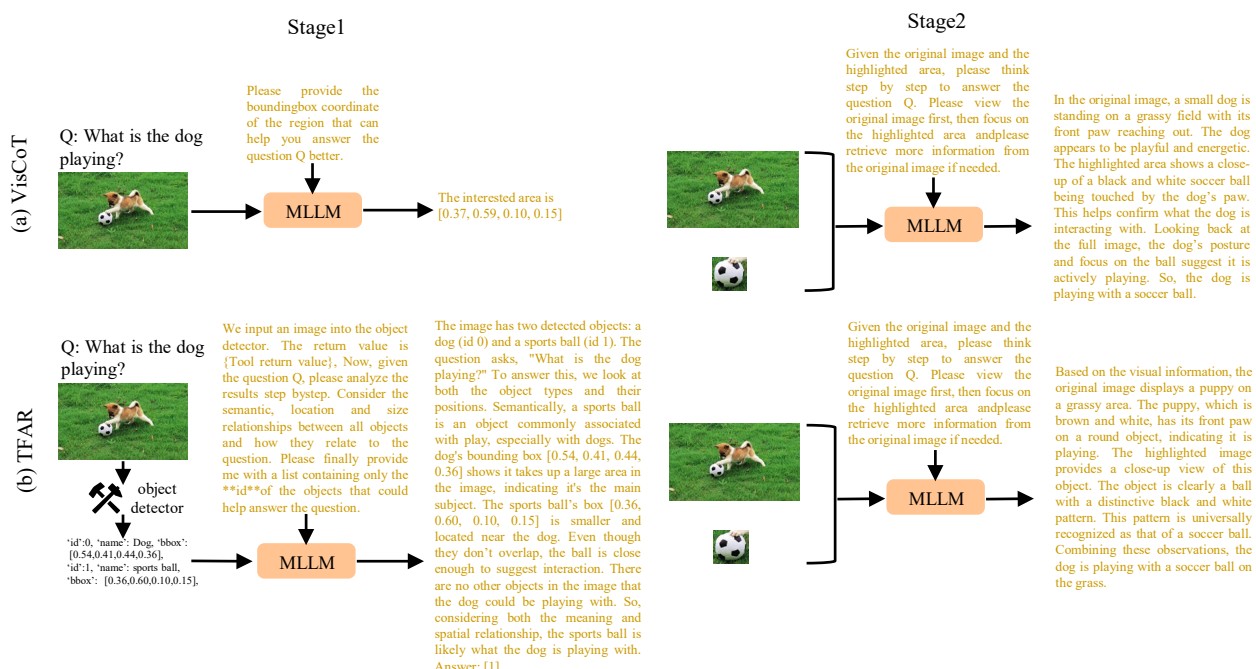

Figure 8: An example illustrating the difference between TFAR and VisCoT in terms of two-stage inference length.

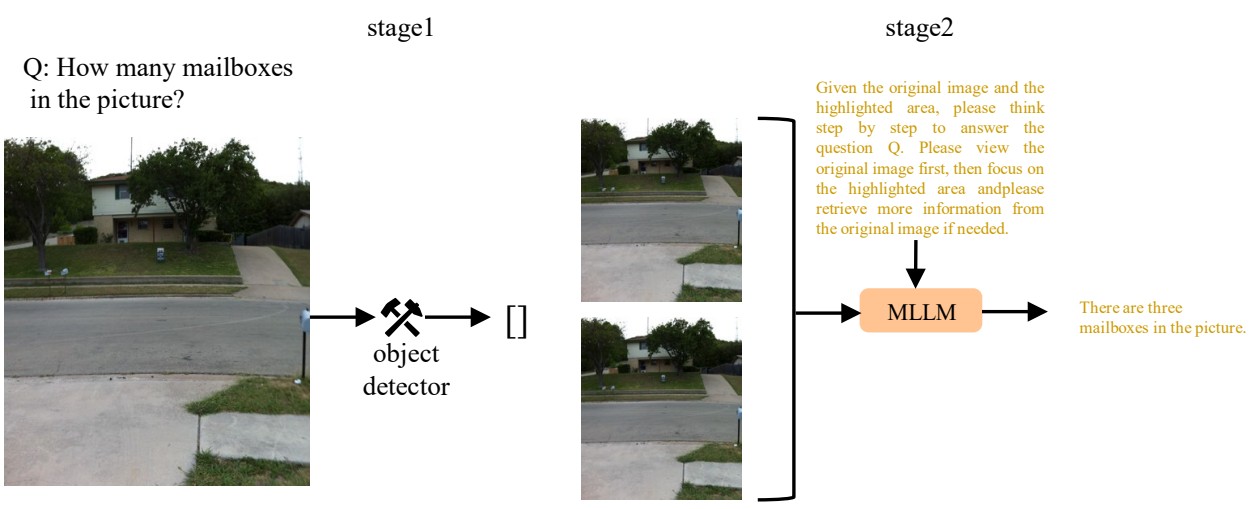

Figure 9: An example of a tool detection failure.

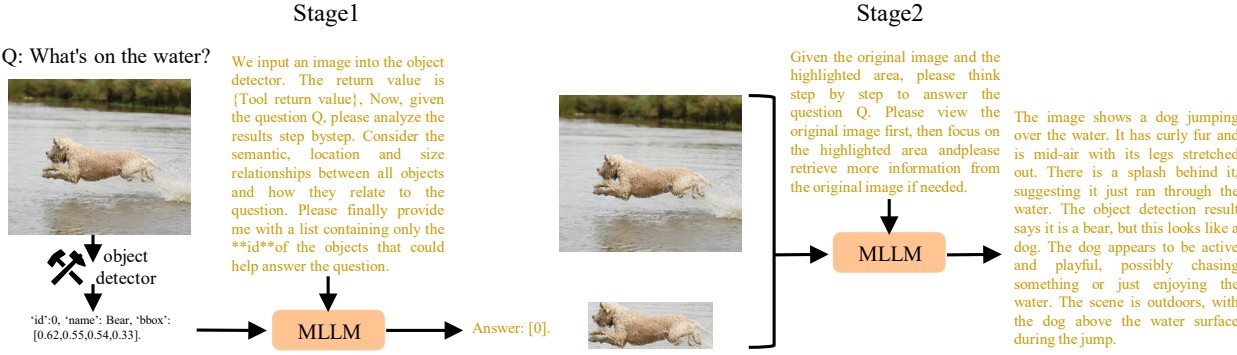

Figure 10: An example of a tool detection wrong.

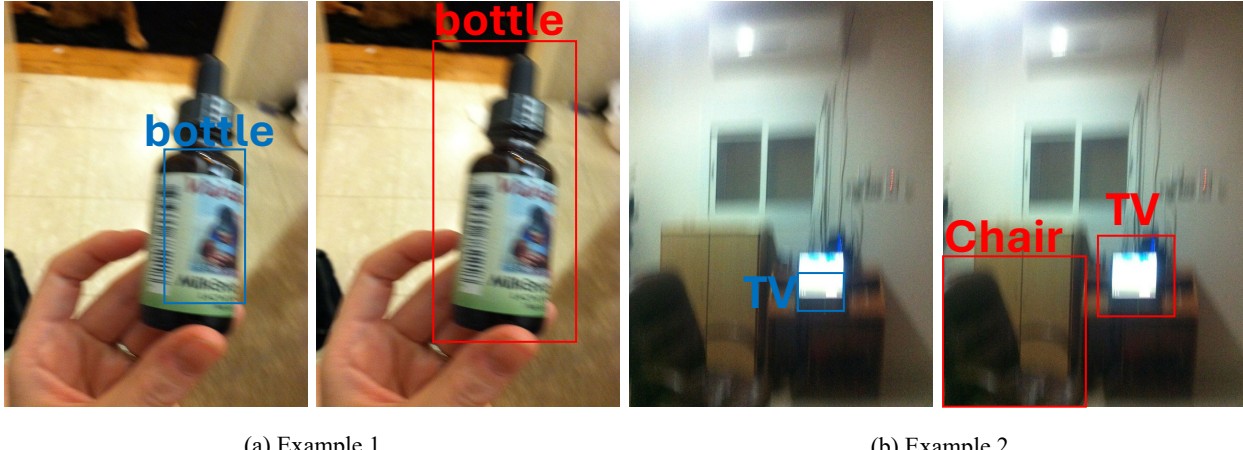

(a) Example 1

(b) Example 2

Figure 11: Some calibrated results from the VizWiz dataset. Images with blue bounding boxes indicate the outputs from the object detection tool, while red bounding boxes represent the calibrated results after applying our correction method.

challenges for machine vision systems. For instance, in Fig. 11(a), only a small part of a bottle was detected by the tool, but our proposed conformal prediction method successfully corrected the bounding box to fully cover the object. Similarly, in Fig. 11(b), the object detection tool failed to detect the chair and produced an incorrect bounding box for the TV. With the help of our correction process, the chair was accurately identified, and a more reasonable bounding box was assigned to the TV.

