# OpenReview forum: "TFAR: A Training-Free Framework for Autonomous Reliable Reasoning in Visual Question Answering"
_TMLR — Accepted by TMLR_

### Review · Reviewer_694L · 2025-07-06

**Summary Of Contributions:**

The paper presents a training free framework for reasoning in Visual QA tasks. The paper introduces:

- Using common lightweight visual tools to provide basic information about the image.
- Introduces conformal prediction-inspired calibration strategy to quantify uncertainty from both external tools and MLLM outputs.

**Audience:**

Yes

**Claims And Evidence:**

Yes

**Requested Changes:**

- The citations in section 2.2 use non-standard style. Like V.K. et al. Kostumov et al. (2024).
- For the ablation involving more tools (TFAR+), the authors use the same tools: Segmentation and Detection. But this ablation should include complementary visual tools like zooming in which can provide more information to the MLLM. Using the same tool just makes it redundant.

**Strengths And Weaknesses:**

Strengths:

- The paper explains everything in a simple manner with examples.
- Gives thorough theoretical explanation of calculating uncertainty for the tools and MLLM output.
- Provides thorough experiments with all possible types of ablations to show the effectiveness of the proposed method.
- The results show improvement over base models and over some fine-tuned models as well.

Weaknesses:
- The inference time is greater for TFAR than fine-tuned models due to tool calling and uncertainty calibration.
- The ablation study for TFAR+ doesn't use any new visual tools.

---

> ### Author Response · Authors · 2025-08-01
> **response to Reviewer 694L**
>
> ***First and foremost, we sincerely thank you for your valuable suggestions. We have carefully reviewed all the comments and revised the manuscript accordingly. The revised or newly added contents in the paper are highlighted in blue. Below, we provide detailed responses to each comment.***
>
> **Weaknesses 1**: The inference time is greater for TFAR than fine-tuned models due to tool calling and uncertainty calibration.
> **Response**: Thank you very much for your insightful comment. We acknowledge that TFAR incurs a higher inference time compared to fine-tuned models, primarily due to the additional steps of tool calling and uncertainty calibration.  TFAR’s inference time is 12.6% and 12.4% slower than the baseline on VQA2 and VizWiz, respectively. However, we believe this overhead remains within a reasonable and practical range—especially when contrasted with the significant computational cost associated with model fine-tuning. For example, LLaVA-ov-7B-VisCoT requires around 20 hours of fine-tuning on 438k question-answer pairs, making it both time-consuming and resource-intensive.
>
> Importantly, the modest increase in inference time yields substantial performance gains: TFAR achieves improvements of 3.5%, 6.5%, 2.5%, 6.1%, and 4.8% across the five datasets evaluated. We believe this demonstrates a favorable trade-off between inference efficiency and accuracy, aligning with recent trends in the literature (lee2025well, hou2025thinkprune), where accuracy gains are often prioritized, even at the cost of slightly longer inference times. Overall, we are confident that the additional inference time is justified by the significant benefits TFAR provides in terms of accuracy and practical deployment flexibility.
>
> **Weaknesses 2**: The ablation study for TFAR+ doesn't use any new visual tools.
> **Response**: Since this question is same with **Requested Changes 2**, please refer to our response there, thanks!
>
> **Requested Changes 1**: The citations in section 2.2 use non-standard style. Like V.K. et al. Kostumov et al. (2024).
> **Response**: Thanks for your  valuable feedback. We have checked all the citations in the paper and revised the incorrect format.
>
> **Requested Changes 2**: For the ablation involving more tools (TFAR+), the authors use the same tools: Segmentation and Detection. But this ablation should include complementary visual tools like zooming in, which can provide more information to the MLLM. Using the same tool just makes it redundant.
> **Response**:  Thank you for your valuable comment. We appreciate the opportunity to clarify our experimental choices. As stated in the Introduction, our primary goal in introducing additional tools is to enable the model to autonomously select regions of interest to guide CoT reasoning. Therefore, our requirements for these tools focus on providing both semantic and positional information of objects in the image. Currently, object detection and segmentation models are best suited for this purpose.
>
> We are grateful for your suggestion to explore alternative tools such as a Zoomin tool. Inspired by this, we conducted additional experiments where, after the MLLM selects objects of interest, we magnified those objects ( in TFAR, we mask out other objects that are not of interest). Specifically, we enlarge each object to the original image size. To avoid overlap among multiple objects of interest, we input all magnified images as separate images alongside the original image into the final MLLM. We refer to this approach as TFAR-zoomin and evaluated it on all datasets. The results are shown in the table below.
>
> **Table: Comparison of different ROI representation methods**
> | Method       | VQA2 | VizWiz | GQA  | Flickr30K | MMBench | Average |
> |--------------|------|--------|------|-----------|---------|---------|
> | **TFAR**     | **82.5** | 60.1  | **67.3** | **67.2**    | **77.1** | **70.8** |
> | TFAR-zoomin  | 82.0 | **60.3** | 66.9 | 66.8      | **77.1** | 70.6    |
>
> Our results indicate that TFAR-zoomin performs slightly better than or comparable to TFAR on some datasets (e.g., VizWiz and MMBench), but overall, its performance is slightly lower than TFAR. We hypothesize that magnifying objects in the middle of the image may disrupt the original spatial relationships, potentially misleading the MLLM's final judgment. Based on these findings, we believe that the original TFAR settings, which maintain spatial consistency, remain preferable.
>
> Once again, we thank you for the insightful suggestion, which helped us further validate and strengthen our methodology. We have added this experiment in the **Section 4.3 Ablation study**, **'Different ROI representation method in TFAR'**.

---

> > ### Comment · Reviewer_694L · 2025-08-04
> >
> > Thank you for the clarifications. I do not have any other issues.

---

> > > ### Author Response · Authors · 2025-08-05
> > >
> > > Thanks again for your time and effort in evaluating our manuscript. Your constructive feedback is greatly appreciated and has helped us improve the quality of our work.

---

### Review · Reviewer_V6XM · 2025-07-13

**Summary Of Contributions:**

This paper introduces a training-free method called TFAR to combine the outputs of bounding box detection tools and image segmentation tools with MLLM Chain-of-Thought reasoning to improve the performance in VQA datasets. First, tools generate segmentation masks or bounding boxes, which are then improved by uncertainty estimation techniques. The outputs are then provided to the MLLM, which selects the appropriate bounding boxes or segmentation masks. These are used to create a second image that only shows the selected regions, and the MLLM uses both images to provide a final answer. Finally, again using uncertainty scores, the answer from the MLLM using bounding boxes or the answer from the MLLM using segmentation masks is selected.

**Audience:**

Yes

**Broader Impact Concerns:**

This paper does not have a Broader Impact Statement. However, while ethical implications such as bias compounding through the pipeline are conceivable, this work does not target any specifically sensitive applications.

**Claims And Evidence:**

Yes

**Requested Changes:**

**(R1)** Formatting: Currently, all citations are using `\citet`, however in many cases, `\citep` is appropriate. These have to be fixed.

**(R2)** Explain how uncertainty estimation is applied to API models such as GPT-4o that do not give access to raw probabilities.

**(R3)** Expand the discussion on inference efficiency: How do the output lengths of TFAR and VisCoT differ? What is the impact of 2 required inference runs of TFAR?

**(R4)** Provide a few examples of failure cases. To which parts in the pipeline can they be traced (tool output, MLLM region selection, final MLLM reasoning)? Where is the greatest potential for improvement?

**Strengths And Weaknesses:**

## Strengths

**(S1)** The proposed method replaces MLLM fine-tuning with tool calling, so it becomes training free. This is a desirable property for many reasons, including efficiency. Furthermore, we generally want to change large models as little as possible, as this might affect performance in other domains.

**(S2)** While the method is mainly designed and evaluated on VQA tasks, the paper clearly points out that TFAR can be extended to other tools. It also appears possible to integrate the method into interactive settings, where models use additional information about images to provide their answer.

**(S3)** Overall, the study is technically solid, and most relevant details are explained in appropriate depth. Results show that TFAR provides benefits, and motivates further research on tool calling as part of reasoning. The paper furthermore motivates research on improving performance of external tools in such scenarios.

## Weaknesses

**(W1)** The main weakness seems to be the reliance on providing two input images and the necessity to perform a separate inference run for each tool considered. This decreases performance and multiple images as input may not be supported by all models.

**(W2)** It is unclear if the discussion on inference efficiency considers the two inference runs required by the proposed method to select the correct tools. More details on where TFAR and VisCoT differ in terms of output length would help understand this better. Furthermore, the inference speed should also be compared to base models with a single inference run, to understand the overhead better.

**(W3)** An interesting aspect to consider is to let the MLLM decide which tool in wants to invoke, if any tool at all. This could potentially decrease inference latency considerably, if the model is confident it knows the correct answer with any tools.

**(W4)** It is unclear how Uncertainty Estimation works for API models such as Gemini-2.5 and GPT-4o, which do not give access to raw probabilities. This should be explained in the paper.

**(W5)** The evaluated open models are outdated at this point. While it would have been great to see experiments on more recent models, at the rate new models are released currently, this may be of minor importance.

**(W6)** One missing aspect is failure cases where the bounding box detection model or segmentation model does not detect the correct image area at all. It would be interesting to see if the MLLM can reject any tool output, labeling it as insufficient on its own.

---

> ### Author Response · Authors · 2025-08-01
> **response to Reviewer V6XM Weekness**
>
> ***First and foremost, we sincerely thank you for your valuable suggestions. We have carefully reviewed all the comments and revised the manuscript accordingly. The revised or newly added contents in the paper are highlighted in blue. Below, we provide detailed responses to each comment.***
>
> **Weaknesses 1**:  The main weakness seems to be the reliance on providing two input images and the necessity to perform a separate inference run for each tool considered, and multiple images as input may not be supported by all models.
> **Response**: Thank you for your thoughtful comment and for highlighting this important aspect of our method. We acknowledge that TFAR requires two input images and a separate inference run for each tool considered, which results in a modest increase in inference time. TFAR’s inference time is 12.6% and 12.4% slower than the baseline on VQA2 and VizWiz, respectively. Nevertheless, we believe this overhead remains within a practical and acceptable range, especially when contrasted with the significant computational cost and resource requirements of large-scale model fine-tuning. For example, LLaVA-ov-7B-VisCoT requires approximately 20 hours of fine-tuning on 438k question-answer pairs, which is both time-consuming and resource-intensive.
>
> Crucially, the slight increase in inference time leads to substantial performance gains: TFAR achieves improvements of 3.5%, 6.5%, 2.5%, 6.1%, and 4.8% across the five datasets evaluated. We believe this represents a favorable trade-off between efficiency and accuracy, which is also reflected in recent literature (lee2025well, hou2025thinkprune), where accuracy improvements are often prioritized even if they incur a modest increase in inference time. Overall, we are confident that the additional inference cost is well justified by the significant accuracy gains and the enhanced flexibility TFAR offers for practical deployment.
>
> Additionally, we would like to point out that supporting multiple image inputs has increasingly become a standard capability of modern MLLMs, as evidenced by recent models such as Qwen2.5-VL and InternVL3 (bai2025qwen2, zhu2025internvl3). This trend will further facilitate the practical adoption of TFAR in the future.
>
> Once again, we appreciate your valuable feedback and the opportunity to clarify the strengths of our approach.
>
> **Weaknesses 2**: It is unclear if the discussion on inference efficiency considers the two inference runs required by the proposed method to select the correct tools. More details on where TFAR and VisCoT differ in terms of output length would help understand this better. Furthermore, the inference speed should also be compared to base models with a single inference run, to understand the overhead better.
> **Response**:  Since this question is same with **Requested Changes 3**, please refer to our response there, thanks!
>
> **Weaknesses 3**:  An interesting aspect to consider is to let the MLLM decide which tool in wants to invoke, if any tool at all. This could potentially decrease inference latency considerably if the model is confident it knows the correct answer with any tools.
> **Response**: Thank you for your valuable comments. This is an important suggestion and aligns with the focus of many agent-based approaches—training MLLMs to select the most appropriate tool for a given task (shi2024learning, haque2025advanced). However, such methods typically require extensive training to achieve accurate tool selection. In contrast, our work aims to enhance the reasoning capabilities of MLLMs in training-free settings. Therefore, we did not include a comparison with these methods in our experiments. Nonetheless, we acknowledge the value of this direction and will consider it in future work. To make this clearer, we have added the above discussion to the **Conclusion** section of the revised paper.
>
> **Weaknesses 4**:  It is unclear how Uncertainty Estimation works for API models.
> **Response**: Since this question is same with **Requested Changes 2**, please refer to our response there, thanks!
>
> **Weaknesses 5**:  While it would have been great to see experiments on more recent models, at the rate new models are released currently, this may be of minor importance.
> **Response**: Thank you for your valuable comments. We have conducted experiments using recent commercial models such as GPT-4o and Gemini 2.5, with the results presented in Table 5 of the paper. Due to time constraints at this stage, we plan to include additional experimental results after acceptance and before submitting the final camera-ready version.
>
> **Weaknesses 6**:  One missing aspect is failure cases where the bounding box detection model or segmentation model does not detect the correct image area at all. It would be interesting to see if the MLLM can reject any tool output, labeling it as insufficient on its own.
> **Response**: Since this question is same with **Requested Changes 4**, please refer to our response there, thanks!

---

> ### Author Response · Authors · 2025-08-01
> **response to Reviewer V6XM Requested Changes**
>
> **Requested Change 1**:  Citation format.
> **Response**: Thank you for your valuable comments. We have carefully reviewed all reference citations throughout the manuscript to ensure that the formatting complies with the required guidelines.
>
> **Requested Change 2**:  Explain how uncertainty estimation is applied to API models.
> **Response**: Thank you for your valuable comments. We apologize for omitting the relevant instructions in the experimental description. You are correct that API-based models such as Gemini-2.5 and GPT-4o do not provide access to raw probability outputs. To address this, we followed a heuristic approach inspired by (zhi2025videoagent2), allowing the LLM to produce its own uncertainty score, in line with the concept of self-reflection (ji2023towards). Specifically, we appended the following instruction to the original prompt: "Note that you need to generate an uncertainty score for your answer, scaled from 1 to 5. The larger the value, the more certain you are." We have added this clarification to the **Section 4.3 Ablation Study**, **'Different base models'**, in the revised version of the paper.
>
> **Requested Change 3**:  Discussion on inference efficiency.
> **Response**: Thank you for your valuable comment. TFAR performs two reasoning paths in parallel during inference, and the reported inference time corresponds to the longest of the two paths. To help compare the difference in inference processes between TFAR and VisCoT, we provide the average number of output tokens in the two reasoning rounds on the VQA2 dataset, as shown below.
>
> **Table: Comparison of output token number in two stages between VisCoT and TFAR**
> | Method  | Stage 1 | Stage 2 |
> |---------|---------|---------|
> | TFAR    | 132     | 115     |
> | VisCoT  | 14      | 132     |
>
> The primary difference in CoT length between TFAR and VisCoT occurs in the first stage. We illustrate this using the example in Fig. 8 in the revised paper. In stage 1, TFAR explicitly prompts the MLLM to consider the semantics, size, and spatial relationships of all detected/segmented objects and select the index of the object of interest. In contrast, VisCoT has been trained on millions of examples to rapidly locate regions of interest, resulting in a shorter CoT in the first round. It explains why TFAR is slightly less efficient during inference.
>
> In stage 2—when generating the final answer based on the original image and the selected RoI—the CoT lengths for both methods are comparable, as they receive similar image inputs and prompts. We have added this analysis to **Appendix C**, **'Comparison of reasoning length between TFAR and VisCoT'**.
>
> Additionally, we have included a comparison of inference time with the base model in the **Section 4.3 Ablation Study**, **'Inference efficiency'**. It is worth noting that the overall inference speed of both TFAR and VisCoT is significantly lower than that of the base model. This slowdown is primarily due to the two-round reasoning framework adopted by both methods to improve answer quality.
>
> **Requested Change 4**:  More failure cases.
> **Response**: Thank you for your suggestion. We agree that presenting more failure cases of tools can help introduce the motivation and effectiveness of TFAR. We add the following content to **Appendix C**, **'Visualization of detection failure and detection error of tools.'** Based on our analysis of the experimental results, we observed that when a tool fails to detect correct information, it is often due to the image being inherently complex, blurry, or ambiguous. In such cases, the MLLM’s own ability to understand the image is also limited.
>
> These failure cases can be categorized into two types:
> (1) Detection failure – where the tool does not return any information. In this situation, we fall back to using the original image as the input region of interest for the MLLM, essentially bypassing the tool. Whether the MLLM can still provide a correct answer depends entirely on its own visual reasoning capability. For example, in Fig. 9 in the revised paper, the object detector failed to identify any objects, yet the MLLM was still able to answer correctly.
> (2) Detection error – where the tool returns incorrect or misleading information. In most of these cases, the MLLM is negatively influenced by the tool’s output, highlighting the importance of tool correction. While we provide examples of such failure cases in Fig.4, 7 and 2 in the revised paper, we also include a success case in which the MLLM correctly rejects the tool's faulty output in Fig. 10 in the revised paper. However, such outcomes are rare. When the tool produces incorrect results, it often indicates that the image is particularly challenging, making it difficult for the MLLM to compensate for the error on its own.
>
> Based on all results and analyses, we believe that the most promising improvement is to employ higher-performing tools, supplemented by a calibration step using a larger validation set.

---

> > ### Comment · Reviewer_V6XM · 2025-08-03
> >
> > Thank you very much for the extensive answer. Most of my comments have been addressed. When reviewing the revision, I still noted the following problems:
> >
> > ---
> >
> > > Non-Standard references
> >
> > The revision still contains at least one non-standard reference ("Z. Z et al."). Please check again carefully all references.
> >
> > > Explain how uncertainty estimation is applied to API models.
> >
> > Thank you for providing this clarification. It is indeed very important to mention this. Additionally, could the paper be revised to add a brief explanation of how to break ties? Given the limited scale (1-5), it is very likely that ties will occur. Furthermore, is it possible to give more details (e.g. in the supplementary results) how the output of API models is parsed? Since in this case models output two pieces of information, the answer and the confidence score.
> >
> > ---
> >
> > I recommend fixing these points to strengthen the paper further.

---

> ### Author Response · Authors · 2025-08-03
> **Response to Reviewer V6XM**
>
> ***Thank you very much for reviewing our revisions and providing new suggestions. We have carefully read these comments and revised the relevant parts in the paper. In addition, we provide detailed responses to each comment below.***
>
> **1. Non-Standard references**
> **Response**: Thank you for your valuable comments. We have carefully reviewed all reference citations again to ensure that the formatting complies with the required guidelines.
>
> **2. Explain how uncertainty estimation is applied to API models.**
> **Response**: Your comments are very insightful! Indeed, we encountered the issue you mentioned when directly applying the method: due to the limited numerical resolution, two paths can yield identical uncertainty scores. To mitigate this, we made a minor practical adjustment, which we regret not explaining in detail due to space limitations. Specifically, we constrained the uncertainty scores output by the MLLM to one decimal place to reduce the likelihood of ties. Furthermore, to facilitate consistent parsing of MLLM responses, we provided a JSON template for the model to follow. The full prompt used for commercial models in Stage 2 is provided below:
>
> *'''
> Given the original image and the highlighted area, please think step by step to generate the answer [ANSWER] of the question Q. Please view the original image first, then focus on the highlighted area and please retrieve more information from the original image if needed.  Additionally, you need to give me [UNCERTAINTY SCORE] that indicates your uncertainty in answering the question accurately, on a scale from 1.0 to 5.0  and accurate to one decimal place. The larger the value, the more certain you are. YOU MUST output in the JSON format:
> {
> 'ANSWER':[ANSWER],
> 'UNCERTAINTY SCORE': [UNCERTAINTY SCORE]
> }
> '''*
>
>  We have added these descriptions in **Appendix  B, 'Prompt for the commercial models in stage 2'**. Thanks!

---

> > ### Comment · Reviewer_V6XM · 2025-08-04
> >
> > Thank you for the fast reply and new revision. I do not have further questions or requested changes.

---

> > > ### Author Response · Authors · 2025-08-04
> > >
> > > Thanks again for your  time and effort in evaluating our manuscript. Your constructive feedback is greatly appreciated and has helped us improve the quality of our work.

---

### Review · Reviewer_2GXx · 2025-07-20

**Summary Of Contributions:**

The paper presents a training-free framework for MLLMs to improve reasoning capabilities for the VQA task. To achieve this, the authors leverage available vision tools, such as scene segmentation and object detection, to obtain regions of interest (RoIs) in the queried images, which they incorporate in the context of MLLMs to enhance reasoning capabilities and improve VQA performance. To ensure the reliability of the vision tools in providing the RoIs, the authors propose a calibration strategy using the conformal prediction for each vision tool in the framework. They provide a specific implementation of their framework, incorporating segmentation and object detection as two vision tools, and using the COCO dataset as the calibration dataset for conformal prediction-based calibration of both tools. Through quantitative experiments, qualitative comparisons, and ablation studies, the authors demonstrate the VQA performance of their proposed implementation of the framework as well as its relevant variants.

**Audience:**

Yes

**Claims And Evidence:**

Yes

**Requested Changes:**

1. Please include details to address the weaknesses mentioned above.

2. Citations in the main text intermix with the text and do not appear to be appropriately punctuated, and impede readability. My recommendation is to use [] for indirect citations in the main text, for example: Sec. 1, Line 1: "such as hallucination Wu et al. (2025)" -> "such as hallucination [Wu et al. (2025)]".

3. In Sec. 4.2, to support the claim that "VizWiz exhibits the largest performance gain, likely because this dataset contains many blurred images", it would be helpful to at least show qualitative examples of improved performance on blurry images.

**Strengths And Weaknesses:**

## Strengths

1. The proposed calibration strategy addresses a key roadblock -- that of making the performance of the individual vision tools in the reasoning framework of MLLM-based VQA reliable -- and thereby makes training-free VQA for MLLMs viable.

2. While demonstrated for VQA, the proposed framework and calibration strategy can be potentially useful for a wide variety of MLLM tasks, such as captioning, summarization, re-identification, and so on.

3. The experimental results are presented in a well-balanced manner, incorporating relevant variants of the proposed implementation of the framework and the calibration strategy, and also clearly delineating the trade-offs in inference efficiency of the proposed training-free approach.

## Weaknesses

1. Sec. 3.2.1 is somewhat hard to follow, and some aspects of the conformal prediction approach can be made clearer:

  - introducing $\alpha$ before mentioning $1 - \alpha$,

  - describing what $X_{test}$ and $Y_{test}$ are in Eqns. 1 and 2,

  - instead of repeating Eqn. 4 in words ("one minus ... "), providing more insight, such as "probability of the model *not* predicting the true class",

  - clarifying the core idea that choosing the quantile as per Eqn. 3 mathematically ensures that the prediction set of labels on the test data contains the true label with probability $1 - \alpha$.

2. Based on Eqn. 9, each pixel label is determined greedily based on the probability vector associated with only that pixel and does not consider the labels of neighboring pixels. How does this approach ensure that the detected segments continuously span the underlying objects? For example, a segment for one object may erroneously contain within it segments of another semantically similar object due to some image or detection noise. Wouldn't such artifacts make the segments unusable for MLLM reasoning?

3. Additionally, why is a non-background label always forced for segmentation? Could this not lead to overcorrecting and detecting false foreground segments?

---

> ### Author Response · Authors · 2025-08-01
> **response to Reviewer 2GXx weakness 1**
>
> ***First and foremost, we sincerely thank you for your valuable suggestions. We have carefully reviewed all the comments and revised the manuscript accordingly. The revised or newly added contents are highlighted in blue. Below, we provide detailed responses to each comment.***
>
> **Weaknesses 1.1**: introducing $\alpha$ before $1-\alpha$.
> **Response**: Thanks for your valuable comment. We agree that we should explain that $\alpha$ represents the allowable error rate (or risk level), before mentioning $1-\alpha$ as the confidence level. We have revised the related part as: Given a pre-trained model and  an allowable error rate (or risk level) $\alpha$, the inductive CP constructs prediction sets that contain the true label with probability at least $1 - \alpha$, where  $1 - \alpha$ is also named as the confidence level.
>
> **Weaknesses 1.2**: describing what $X_{test}$ and  $Y_{test}$ are in Eqns. 1 and 2.
> **Response**: Thanks for your valuable comment, we agree that $X_{\text{test}}$ and $Y_{\text{test}}$ should be described clearly. We have revised the related part and described the details of $X_{\text{test}}$ and $Y_{\text{test}}$ as follows:
>
> In classification, one chooses a *nonconformity score* $s(\cdot)$ to measure how “unusual” each predicted label is (relative to the ground truth), then estimates a threshold $\hat{q}_\alpha$ from a held-out *calibration set*.
>
> Formally, let  ${(X_i, Y_i)}, i = 1, ..., n$ denote a calibration set of size $n$, where $X_i$ is an input sample and $Y_i$ is its true label. Given a new, previously unseen test input $X_{\text{test}}$ and its (unknown) true label $Y_{\text{test}}$, conformal prediction produces sets:
>
> $$
> \mathcal{C}(X_{\text{test}}) = \{ \hat{y} \;|\; s(X_{\text{test}}, \hat{y}) \le \hat{q}_\alpha \}
> $$
>
> subject to:
>
> $$
> \mathbb{P}(Y_{\text{test}} \in \mathcal{C}(X_{\text{test}})) \ge 1 - \alpha
> $$
>
> where $X_{\text{test}}$ denotes the test input and $Y_{\text{test}}$ denotes its (unknown) ground-truth label.
>
> **Weaknesses 1.3**: instead of repeating Eqn. 4 in words ("one minus ... "), providing more insight, such as "probability of the model not predicting the true class".
> **Response**: Thanks for your valuable comment, we agree that we have revised the related part and changed the explanation of the Eqn. 4 as:  For a classification model whose output is a probability vector over classes, a common choice of nonconformity score is
> $$
>     s(X_i, Y_i) =1 - p(Y_i \mid X_i\bigr),
> $$
> i.e., the probability of the model not predicting the true class.
>
> **Weaknesses 1.4**: clarifying the core idea that choosing the quantile as per Eqn. 3 mathematically ensures that the prediction set of labels on the test data contains the true label with probability $1-\alpha$.
> **Response**:  Thanks for your valuable comment, we agree that it is necessary to emphasize here that Eqn. 3  is the core of the CP method, which will help readers understand our research motivation. We have revised the related part as: The core principle of CP is that choosing the quantile as shown in  Equation 3 mathematically ensures that the prediction set of labels on the test data contains the true label with probability  $1-\alpha$.

---

> ### Author Response · Authors · 2025-08-01
> **response to Reviewer 2GXx weakness 2**
>
> ***First and foremost, we sincerely thank you for your valuable suggestions. We have carefully reviewed all the comments and revised the manuscript accordingly. The revised or newly added contents are highlighted in blue. Below, we provide detailed responses to each comment.***
>
> **Weaknesses 2**: Based on Eqn. 9, each pixel label is determined greedily based on the probability vector associated with only that pixel and does not consider the labels of neighboring pixels. How does this approach ensure that the detected segments continuously span the underlying objects? For example, a segment for one object may erroneously contain within it segments of another semantically similar object due to some image or detection noise. Wouldn't such artifacts make the segments unusable for MLLM reasoning?
> **Response**:  Thank you for this thoughtful question. We appreciate the opportunity to clarify how our conformal prediction-based calibration preserves the spatial coherence and usability of segmentation outputs. Please find our response below:
>
> 1. **The role and mechanics of Equation (9).**
>    Equation (9) is intended to adjust the outputs of existing, pre-trained segmentation models rather than to generate segments independently. As many segmentation models tend to under-segment at boundaries (often misclassifying foreground pixels as background) [xie2021segformer], our approach uses conformal prediction to recover such missed regions.
>    For each pixel, we construct a conformal prediction set $\mathcal{C}(u,v)$ according to a data-driven uncertainty threshold $\hat{q}_\alpha$.
>
>    - If the background class (0) is **not** in $\mathcal{C}(u,v)$, we assign the most likely foreground class in the set.
>    - If the background class is included, we choose the most probable non-background class within the set.
>
>    This targeted relabeling helps recover object parts that may have been misclassified as background, producing more accurate results.
>
> 2. **The segment continuity and integrity.**
>    Although our method operates on pixels individually, it is applied as a post-processing step to the outputs of strong segmentation models (e.g., SEEM), which already incorporate powerful spatial priors on object shapes and continuity. As a result, the segments are generally spatially coherent prior to calibration, and our correction preserves this structure rather than fragmenting it. In addition, recent work (*Chen et al., ICCV 2025, "ConformalSAM"*) demonstrates that such pixel-wise conformal prediction atop robust segmentation models further improves segmentation quality and mask completeness, outperforming more elaborate aggregation schemes.
>
> 3. **The robustness to artifacts and suitability for MLLM reasoning.**
>    Concerns about segments incorporating regions of semantically similar objects usually arise from limitations in the base segmentation model, rather than the calibration step itself. Our conformal prediction calibration is designed to accurately reflect model uncertainty by including plausible classes when ambiguity exists, rather than making overconfident assignments. In cases of high uncertainty, the prediction set may be empty, clearly signaling ambiguity to the downstream MLLM. Empirically, as shown in Figure 3 in *ConformalSAM*, this approach does not introduce additional artifacts or noise, and in practice, it yields more complete and spatially coherent segmentations. This directly benefits the reasoning performance of MLLMs.
>
> ---
>
> In summary, our method leverages the spatial priors of advanced segmentation models and enhances their outputs via a principled, uncertainty-aware correction. This combination leads to more complete and reliable RoIs, as supported by recent empirical studies.

---

> ### Author Response · Authors · 2025-08-01
> **response to Reviewer 2GXx weakness 3**
>
> ***First and foremost, we sincerely thank you for your valuable suggestions. We have carefully reviewed all the comments and revised the manuscript accordingly. The revised or newly added contents are highlighted in blue. Below, we provide detailed responses to each comment.***
>
> **Weaknesses 3**: Additionally, why is a non-background label always forced for segmentation? Could this not lead to overcorrecting and detecting false foreground segments?
> **Response**:   Thank you for your thoughtful follow-up question. We appreciate the opportunity to clarify the safeguards in our calibration approach that prevent overcorrection and the introduction of false foreground segments. Our response is as follows:
>
> 1. **We perform the conditional rather than absolute relabeling.**
>    A key aspect of our method is that a non-background label is not always enforced. The correction is conditional and only activated in specific cases of ambiguity. The logic in Equation (9), which prioritizes a foreground class, is triggered only when a pixel’s conformal prediction set $\mathcal{C}(u,v)$ contains both the background class and at least one plausible foreground class.
>
>    If a pixel is confidently predicted as background—meaning the background class is the sole member of the prediction set, or no foreground class satisfies the confidence threshold $\hat{q}_{\alpha}$—it remains labeled as background. Our method does not generate foreground objects arbitrarily; it resolves ambiguity in favor of a foreground label only when statistically justified.
>
> 2. **The motivation of this design.**
>    This design choice is intended to correct a well-known bias in modern segmentation models: the tendency to under-segment objects by misclassifying true foreground pixels as background, especially near object boundaries (Xie et al., 2021). In downstream VQA tasks, such false negatives can significantly impact the completeness and usefulness of RoIs for MLLM reasoning. Our conditional relabeling targets these specific cases. While we recognize that overcorrection is theoretically possible, the use of a data-driven threshold $\hat{q}_\alpha$ ensures that relabeling is statistically controlled, striking a balance between recovering foreground information and avoiding spurious segments.
>
> 3. **The alignment with the related work.**
>    Our approach is further supported by recent research in uncertainty quantification for segmentation. The work by (*Chen et al., ICCV 2025, "ConformalSAM"*) employs the same strategy, noting that class-conditional filtering is necessary to address background dominance and to favor minority (foreground) classes. The significant performance gains they report, and their visualizations (e.g., Figure 3 in *"ConformalSAM"*), show that their method corrects under-segmentation to produce more complete masks without introducing spurious foreground artifacts.
>
> ---
>
> In summary, our method provides a conditional, data-driven correction for under-segmentation, governed by statistically principled thresholds. This approach is supported by recent literature and best practices for handling class imbalance in segmentation. While the risk of overcorrection exists in theory, our design and empirical results show that the method reliably produces more complete and accurate RoIs for downstream MLLM reasoning.

---

> ### Author Response · Authors · 2025-08-01
> **response to Reviewer 2GXx weakness Requested Changes:**
>
> ***First and foremost, we sincerely thank you for your valuable suggestions. We have carefully reviewed all the comments and revised the manuscript accordingly. The revised or newly added contents are highlighted in blue. Below, we provide detailed responses to each comment.***
>
> **Requested Change 1**:  Please include details to address the weaknesses mentioned above.
> **Response**:  Thanks for your valuable comment. We have checked all the weaknesses and responded to each comment above.
>
> **Requested Change 2**:  Citations in the main text intermix with the text and do not appear to be appropriately punctuated, and impede readability. My recommendation is to use [] for indirect citations in the main text, for example: Sec. 1, Line 1: "such as hallucination Wu et al. (2025)" -> "such as hallucination [Wu et al. (2025)]".
> **Response**:  Thank you for your valuable comment. We have carefully reviewed all the citations in the paper and corrected any inaccuracies to ensure that the citation format fully complies with the required standards.
>
> **Requested Change 3**:  In Sec. 4.2, to support the claim that "VizWiz exhibits the largest performance gain, likely because this dataset contains many blurred images", it would be helpful to at least show qualitative examples of improved performance on blurry images.
> **Response**:  Thank you for your valuable comment. The VizWiz dataset simulates images captured from the perspective of visually impaired individuals and primarily contains blurry or low-quality images. MLLMs generally exhibit limited robustness when processing such inputs directly. In TFAR, dedicated object detection and segmentation tools can effectively assist in identifying these challenging regions, thereby improving overall accuracy. To better illustrate these results, we have added several examples to  **Appendix C, 'Some calibrated results from VizWiz dataset.'**. As shown in Fig. 11 in the revised paper, the VizWiz dataset is characterized by frequent motion blur and poor image quality, which present significant challenges for machine vision systems. For instance, in  Fig. 11(a), only a small part of a bottle was detected by the tool, but our proposed conformal prediction method successfully corrected the bounding box to fully cover the object. Similarly, in  Fig. 11(b), the object detection tool failed to detect the chair and produced an incorrect bounding box for the TV. With the help of our correction process, the chair was accurately identified, and a more reasonable bounding box was assigned to the TV.

---

> ### Comment · Reviewer_2GXx · 2025-08-05
> **Response to Authors**
>
> I thank the authors for their detailed responses, which address my questions and concerns. I do not have further questions.

---

> > ### Author Response · Authors · 2025-08-06
> >
> > Thanks again for your time and effort in evaluating our manuscript. Your constructive feedback is greatly appreciated and has helped us improve the quality of our work.

---

### Decision · Action_Editor_R4Jz · 2025-08-25

**Recommendation:** Accept as is

**Additional Comments:**

The paper introduces TFAR, a Training-Free Framework for Autonomous Reliable Reasoning in VQA. The approach leverages external tools, such as image segmentation models and object detectors, to guide the reasoning process, and augments them with an Uncertainty Quantification (UQ) measure based on the conformal prediction formalism. The MLLM then takes as input the image, enhanced with localized visual cues and their associated uncertainty scores, and produces answers for VQA tasks. This method outperforms a vanilla MLLM and achieves results comparable to models fine-tuned with fine-grained labels, while adding only a reasonable overhead in inference time.

The paper initially received positive reviews, with reviewers acknowledging the relevance of the idea and the strength of the experiments. However, they also requested clarifications regarding the UQ module (e.g., its application to API models, formalization in the submission) and discussions of the method's efficiency. During the discussion period, the authors provided adequate clarifications, and all reviewers ultimately recommended acceptance.

The AE has carefully reviewed the submission and the discussion, and acknowledges that the paper makes a valuable contribution to the challenge of reasoning with MLLMs in VQA. The AE therefore recommends acceptance.

**Audience:**

Yes

**Audience Explanation:**

The paper addresses the challenge of reasoning with multimodal large language models (MLLMs) in Visual Question Answering (VQA)—a highly relevant topic in today’s AI landscape. It is likely to be of significant interest to a wide TMLR audience.

**Claims And Evidence:**

Yes

**Claims Explanation:**

This is an interesting paper showing the viability of using external vision tools to improve the reasoning capabilities of multimodal large language models (MLLMs) in Visual Question Answering (VQA).

---

> ### Author Response · Authors · 2025-08-28
>
> We would like to sincerely thank the action editor and reviewers for their constructive feedback and thoughtful suggestions, which have greatly improved the quality and clarity of this paper.